# Inferring genetic interactions from comparative fitness data

**Kristina Crona[1†], Alex Gavryushkin[2,3†], Devin Greene[1†], Niko Beerenwinkel[2,3]***

[1]Department of Mathematics and Statistics, American University, Washington, DC, United States; [2]Department of Biosystems Science and Engineering, ETH Zurich, Basel, Switzerland; [3]SIB Swiss Institute of Bioinformatics, Basel, Switzerland

**Abstract** Darwinian fitness is a central concept in evolutionary biology. In practice, however, it is hardly possible to measure fitness for all genotypes in a natural population. Here, we present quantitative tools to make inferences about epistatic gene interactions when the fitness landscape is only incompletely determined due to imprecise measurements or missing observations. We demonstrate that genetic interactions can often be inferred from fitness rank orders, where all genotypes are ordered according to fitness, and even from partial fitness orders. We provide a complete characterization of rank orders that imply higher order epistasis. Our theory applies to all common types of gene interactions and facilitates comprehensive investigations of diverse genetic interactions. We analyzed various genetic systems comprising HIV-1, the malaria-causing parasite *Plasmodium vivax*, the fungus *Aspergillus niger*, and the TEM-family of $\beta$-lactamase associated with antibiotic resistance. For all systems, our approach revealed higher order interactions among mutations.

DOI: https://doi.org/10.7554/eLife.28629.001

**\*For correspondence:**
niko.beerenwinkel@bsse.ethz.ch

[†]These authors contributed equally to this work

**Competing interests:** The authors declare that no competing interests exist.

## Introduction

The fitness of an individual with a particular genotype is a measure of its expected contribution to the next generation of the population. The collection of all fitness values for all genotypes, referred to as the fitness landscape, is a central concept in evolutionary biology (*Wright, 1932*; *Orr, 2009*). The fitness landscape can have a strong impact on the fate of the evolving population, such as, for example, the risk of a pathogen population to develop drug resistance and to survive under drug treatment (*de Visser and Krug, 2014*).

Genetic interactions, or epistasis, are abundant in nature. They can have many causes and occur at various scales, for instance, among mutations of a protein-coding sequence or between sequences coding for different genes. Unless there are genetic interactions, we assume that fitness is additive, that is, the fitness effects of individual mutations sum. An additive fitness landscape is determined by the wild-type and single-mutant fitness values.

If the fitness landscape is determined by the wild-type, single-mutant, and double-mutant fitness values, then we say that it has no higher order epistasis. Intuitively, higher order epistasis means that the fitness of a multiple mutant is unexpected given the fitness of the wild type and all single and double mutants. For example, *Weinreich et al. (2006)* showed that five mutations jointly increase antibiotic resistance considerably more than expected.

Measuring fitness experimentally is challenging. Fitness measurements tend to come with high uncertainty and they are often obtained only for a subset of genotypes. Moreover, fitness can sometimes not be measured directly at all. Instead, phenotypes are considered that can be measured and are believed to approximate fitness well. For instance, antimicrobial drug resistance is the dominating survival factor for a bacterial population under drug exposure, so that the degree of resistance is a good substitute measure of fitness. Several such fitness proxies are used in microbiology, including

survival as measured by disc diffusion tests. Although it is possible to study epistasis of the proxy data, in general, presence or absence of epistasis in the proxy landscape does not imply presence or absence of epistasis in the fitness landscape itself.

Experimentally, epistatic interactions have been measured in several genetic systems, including *E. coli* (*Khan et al., 2011*; *Weinreich et al., 2006*; *Poelwijk et al., 2007*), HIV-1 (*da Silva et al., 2010*; *Segal et al., 2004*), and other viruses (*Wylie and Shakhnovich, 2011*; *Sanjuán, 2010*). These and similar studies involve the analysis of standing genetic variation or spontaneous mutations (*Bonhoeffer et al., 2004*; *Bershtein et al., 2006*), engineered site-directed mutations (*Sanjuán et al., 2004*; *Weinreich et al., 2006*), and combinations of both (*Sanjuán et al., 2005*; *Poon and Chao, 2006*). Competition experiments are also frequently employed to learn mutational fitness effects. For example, *Sanjuán et al. (2004)* studied the distribution of deleterious mutational effects in RNA viruses using this approach. Such experiments are typically run on single-nucleotide substitution mutants produced by site-directed mutagenesis. The data produced in competition experiments is informative about pairwise comparisons of genotypes with respect to their fitness. However, little is known about whether or not it is possible to learn higher order genetic interactions from such fitness comparison data.

Due to the rapid growth of the number of possible interactions with the number of loci, all interactions can exhaustively be studied only for a small number of loci. At the human genome scale, for example, a complete study of only pairwise gene interactions would already require hundreds of millions of experiments. On the other hand, for smaller organisms, such as yeast, all pairwise and several three-way gene interactions have been measured experimentally (*Costanzo et al., 2010*). Only when restricting to a small set of preselected loci, can one assess all combinations of mutations and hence all epistatic interactions. This approach has been pursued, for example, by *Weinreich et al. (2006)* for a five-locus system associated with bacterial drug resistance.

Historically, the study of genetic interactions was mostly restricted to pairwise epistasis. According to *Crow and Kimura, 1970* (p. 224) higher order interactions were generally believed not to be significant in nature, with references to Fisher, Haldane, and Wright. More recent arguments for the same view have been stated in the context of protein folding (*Gupta and Adami, 2016*). On the other hand, empirical findings suggest that the opposite is true for many other systems (*Szendro et al., 2013*; *Neidhart et al., 2013*; *Sailer and Harms, 2017a*). For example, *Weinreich et al. (2013)* argue that three-way and four-way interactions can be as strong as pairwise epistasis referring to various empirical fitness studies, and *Knies et al. (2017)* find many epistatic interactions in a numerically near-additive fitness landscape, reducing dramatically the number of accessible evolutionary trajectories. Although the significance of higher order interactions may vary between systems, the topic has not been thoroughly investigated. This is partly due to lack of adequate methodology to quantitatively assess the interactions underlying an observed empirical fitness landscape. Improved mathematical and statistical tools for detecting higher order interactions, as well as more empirical results, are necessary for more conclusive answers regarding the importance of higher order interactions.

In this paper, we consider fitness data that comes in the form of pairwise comparisons. Such data are frequent in practice and can arise in different ways. First, some assays rely on comparing the fitness of two genotypes, for example, by letting them grow in direct competition. Each competition experiment is informative about which of the two genotypes has higher fitness, without estimating the fitness values themselves. Second, direct but uncertain fitness measurements are also often summarized as pairwise fitness relations by recording only whether two genotypes displayed significantly different fitness values or not. Third, rather than fitness itself, a fitness proxy, that is, a phenotype closely related to fitness, may be considered. Fitness proxies cannot be used directly to measure epistasis, because they generally do not preserve fitness linearity (*Gong et al., 2013*), but if proxy data preserves pairwise comparisons, they may be used instead. Lists of mutants found in a new environment, such as, for example, a new host for a pathogen or a drug environment can be utilized similarly. Assuming that the capability to transition to and survive in the new environment is an indication of higher fitness, this type of observational data also provides pairwise fitness comparisons. Similarly, the population frequency of genotypes can sometimes be used to draw conclusions about fitness. For example, by employing a specific model of viral evolution, fitness was inferred computationally from deep sequencing data of an HIV-1 population, and pairwise credible fitness differences were reported (*Seifert et al., 2015*).

Irrespective of how they were obtained, any consistent set of pairwise fitness relations can be regarded as a partial order of the genotypes with respect to fitness. Two specific types of partial orders play important roles for fitness landscapes. First, if comparisons are available for all pairs of genotypes, then the partial order is a total order, or rank order. In this case, all genotypes are ordered according to fitness. Second, several studies compare fitness only between mutational neighbors, that is, genotypes which differ at exactly one locus. The resulting partial orders are referred to as fitness graphs and have recently been used extensively (*Ogbunugafor et al., 2016*; *Wu et al., 2016*; *Smith and Cobey, 2016*; *Mira et al., 2015*).

The question addressed in the present study is whether higher order interactions can be inferred from rank orders, fitness graphs, and general partial orders. Connections between rank orders and fitness graphs to epistasis and global properties of fitness landscapes have been observed repeatedly (*Greene and Crona, 2014*; *Crona et al., 2013*; *Poelwijk et al., 2011*; *Weinreich et al., 2006*; *Weinreich et al., 2005*). Most recently, *Wu et al. (2016)* discussed an example of a fitness graph that implies higher order epistasis. The significance of rank orders of genotypes for epistasis was recognized by *Weinreich et al. (2005)*. The authors introduced the concept of sign epistasis. By definition, a system has sign epistasis if the sign of the effect of a mutation, whether positive or negative, depends on genetic background. Importantly, sign epistasis implies that the rank order of the genotypes is not compatible with additive fitness. In this paper, we develop a related approach based on rank orders that applies to higher order epistasis as well as other measures of gene interactions. For instance, if the rank order of the genotypes implies three-way epistasis for a three-locus system, then one has a signed variant of three-way epistasis. Similarly, one can consider signed variants of almost any type of gene interaction. The theory of sign epistasis stands as a model for development in the area, and there is a potential for understanding global properties of fitness landscapes in terms of (local) signed interactions, similar to results for sign epistasis.

In addition to the theoretical work mentioned above, rank order arguments have been used for developing antimicrobial treatment strategies (*Smith and Cobey, 2016*; *Nichol et al., 2015*; *Mira et al., 2015*; *Goulart et al., 2013*). However, the full potential of rank order consideration for the comprehensive analysis of epistatic gene interactions in general $n$-locus genetic systems has not been exploited. Furthermore, to the best of our knowledge the general case of arbitrary partial fitness orders has yet to be considered.

Here, we develop quantitative tools to detect any type of gene interactions measured by linear forms, including epistasis as described by Fourier coefficients, Walsh coefficients, and circuits (*Beerenwinkel et al., 2007*; *Weinreich et al., 2013*). In particular, our approach applies to total $n$-way epistasis, conditional epistasis, and marginal epistasis. We used our approach to analyze genetic interactions in HIV-1, the parasite *Plasmodium vivax*, the fungus *Asbergillus niger*, and $\beta$-lactamase antibiotic resistance systems. In all cases, we detect higher order interactions based only on partial information about the fitness order of genotypes, without knowing or estimating the actual fitness values.

## Results

We consider genetic systems consisting of $n$ biallelic loci. A genotype can then be represented as a binary string of zeros and ones of length $n$, where $0$ denotes the wild-type allele and $1$ the alternative allele. We assume that fitness is additive in the absence of epistasis. The fitness of a genotype $g$ is denoted by $w_g$, and we assume that the fitness landscape $w$ is generic in the sense that no two genotypes have exactly the same fitness.

A complete analysis of all epistatic interactions would require fitness measurements of all $2^n$ genotypes. However, this level of completeness is rarely available in empirical data sets due to experimental design or an infeasible number of genotypes. To address this limitation, we developed methods that are applicable to partial orders of genotypes according to fitness. For example, the two fitness relations $w_{01} > w_{00}$ and $w_{10} > w_{11}$ together define a partial order. One can always extend a partial order to a rank order, that is, a total order of the genotypes in the system from highest to lowest fitness. For example, the total order $w_{10} > w_{11} > w_{01} > w_{00}$ extends the partial order above. Our goal is to understand what fitness rank orders and more generally partial fitness orders of genotypes reveal about gene interactions.

## Two-locus case

We first consider epistasis for a biallelic two-locus population consisting of the unmutated genotype, or wild type, 00, the two single mutants 01 and 10, and the double mutant 11. In this case, epistasis is denoted by $\varepsilon_2$, where the index two refers to the number of loci. It is defined as the deviation from additivity,

$$\varepsilon_2 = (w_{00} + w_{11}) - (w_{01} + w_{10}). \tag{1}$$

The system has no epistasis if $\varepsilon_2 = 0$, positive epistasis if $\varepsilon_2 > 0$ and negative epistasis if $\varepsilon_2 < 0$.

We first assume that the available information on fitness is a rank order of the genotypes (*Figure 1*). The rank order is sometimes sufficient for determining that the system has epistasis. For instance, the rank order $w_{11} > w_{00} > w_{10} > w_{01}$ (*Figure 1* rank order 3), implies $w_{00} + w_{11} > w_{01} + w_{10}$, so $\varepsilon_2 > 0$. It follows that the rank order alone allows one to detect positive epistasis without knowledge of the actual fitness values. There are 24 rank orders of the biallelic two-locus system. Among these, eight imply positive epistasis, eight imply negative epistasis, and eight do not permit any inference regarding epistasis. In total two thirds of the rank orders imply epistasis. Each rank order that implies epistasis also determines the sign of $\varepsilon_2$ (*Figure 1*).

Sometimes even a partial order of the genotypes is sufficient for determining that the system has epistasis. For instance, if we know that $w_{01} > w_{00}$ and $w_{10} > w_{11}$, then we can infer that the system has negative epistasis (*Figure 2a*). To see this, we consider all rank orders that extend the partial order. There are six such total extensions, namely rank orders 9, 10, 12, 13, 14, and 16 in *Figure 1*, and all imply negative epistasis. We conclude that the partial order implies epistasis, based on only two fitness comparisons and without knowing any of the actual fitness values. This observation holds in general: If all total extensions imply epistasis then the same is true for the partial order. We will use this argument repeatedly.

A partial order can also be compatible with several rank orders, some of which might imply epistasis while others do not. In this case, the information is not sufficient to detect epistasis from the partial order alone. For example, the partial order $w_{00} > w_{01} > w_{10}, w_{11}$ is compatible with the two rank orders $w_{00} > w_{01} > w_{11} > w_{10}$ and $w_{00} > w_{01} > w_{10} > w_{11}$ (*Figure 2b*). The first rank order implies positive epistasis, but the other one does not. Consequently, the partial order does not reveal whether or not the system has epistasis, and further comparisons are needed for a conclusion. A more detailed treatment of partial fitness orders can be found in *Lienkaemper et al. (2017)*.

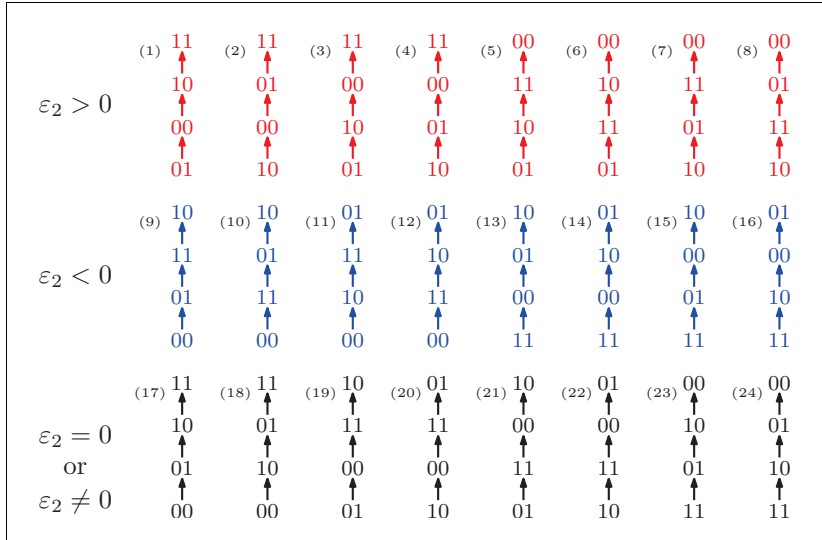

**Figure 1.** All 24 rank orders of the biallelic two-locus system, where the 16 colored rank orders imply epistasis. Red (top row) indicates positive epistasis and blue (middle row) negative epistasis.
DOI: https://doi.org/10.7554/eLife.28629.002

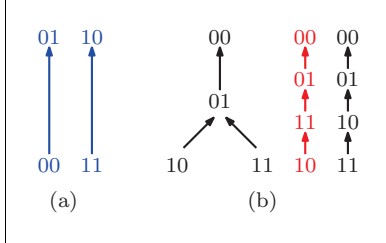

**Figure 2.** (a) A partial fitness order of genotypes. The rank orders that extend this partial order are orders (9 , 10 , 12 , 13 , 14), and (16) in *Figure 1*. All of them imply negative epistasis ($\varepsilon_2<0$). (b) A partial order of genotypes with all its total extensions shown on the right. The first extension shown in red implies positive epistasis ($\varepsilon_2>0$), while the second one in black does not.

DOI: https://doi.org/10.7554/eLife.28629.003

Fitness graphs constitute an important subclass of partial orders, as they often are the reported result of experiments, and because of their relevance for evolutionary processes (*Figure 3*). Briefly, the nodes of a fitness graph represent genotypes and for each pair of mutational neighbors, that is, genotypes which differ at exactly one locus, an arrow points toward the genotype of higher fitness (Section Partial orders and fitness graphs).

A fitness graph implies epistasis exactly when all rank orders compatible with the graph do, as is the case for partial orders in general. For example, *Figure 3* shows the four fitness graphs where genotype 00 has lowest fitness in the in the system. The graphs (b), (c) and (d) imply epistasis, whereas (a) is compatible with additive fitness.

A couple of observations from *Figure 3* are useful for determining if a system is compatible with additive fitness. First, any rank order compatible with the graph (a) has the following property: For each genotype, replacing 0 by 1 results in a genotype of higher fitness. If the genotype 00 has minimal fitness in the system, then rank orders are compatible with graph (a) exactly if they satisfy this property. The second observation is that a fitness graph where 00 has minimal fitness is compatible with additive fitness exactly if all arrows point up. Both observations generalize to any number of loci, and can be phrased in full generality (one can reduce the general problem to the case when $0\ldots0$ has lowest fitness in the system by a relabeling argument). In particular, only 384 out of $(2^3)! = 40,320$ rank orders are compatible with fitness graphs with all arrows up (after relabeling) for the three-locus system, which we consider next (Materials and methods, Section Partial orders and fitness graphs).

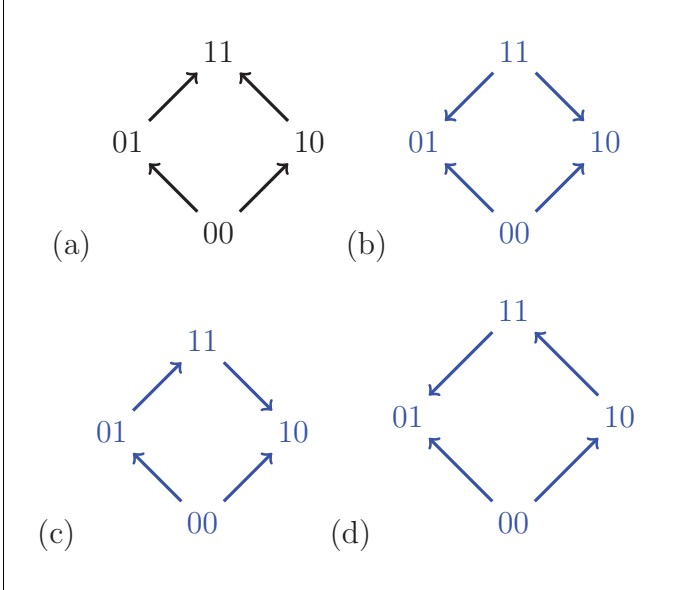

**Figure 3.** For a biallelic two-locus system where the genotype 00 has the lowest fitness, there are four fitness graphs. The graph (a) is compatible with additive fitness, whereas the remaining graphs imply negative epistasis.

DOI: https://doi.org/10.7554/eLife.28629.004

## Three-locus case

The biallelic three-locus system consists of the eight genotypes 000, 001, 010, 011, 100, 101, 110, and 111. The system has total three-way epistasis if

$$\varepsilon_3 = (w_{000} + w_{011} + w_{101} + w_{110}) - (w_{001} + w_{010} + w_{100} + w_{111}) \neq 0. \tag{2}$$

For the three-locus system, we distinguish between fitness landscapes with no epistasis (fitness is additive), with pairwise but not higher order epistasis (fitness is not additive but $\varepsilon_3 = 0$), and with three-way epistasis ($\varepsilon_3 \neq 0$).

Some rank orders imply three-way epistasis, similar to our observation of epistasis in the two-locus case. The condition for when a rank order implies three-way epistasis is remarkably simple, and we demonstrate it with an example. Consider the rank order

$$w_{110} > w_{111} > w_{101} > w_{011} > w_{100} > w_{010} > w_{000} > w_{001}. \tag{3}$$

We can represent this rank order by a word in the letters $e$ and $o$ using the following procedure. The genotype 110 with the highest fitness is represented by $e$ because it has an even number of 1's, the genotype 111 with the second highest fitness is represented by $o$ because it has an odd number of 1's, and so forth. Working from highest to lowest fitness, we obtain the word

$$eoeeooeo. \tag{4}$$

If one reads the word letter by letter from left to right, then one has never encountered more $o$'s than $e$'s. This property means that $eoeeooeo$ is a Dyck word (**Stanley, 1999**).

For a biallelic three-locus system, a rank order implies three-way epistasis exactly if its associated word (where the role of $e$ and $o$ can be interchanged) is a Dyck word (Proposition 1). This simple rule allows us to conclude that an empirical system has three-way epistasis. As in the two-locus case, a landscape may have three-way epistasis even if the rank order does not imply it. For biallelic three-locus systems, there are in total 40,320 rank orders, of which 16,128 (40%) imply three-way epistasis (Proposition 1).

Fitness graphs can be analyzed by using our results on rank orders as in the two-locus case. **Figure 4** shows three fitness graphs for three-locus systems. The fitness graph (a) implies three-way epistasis, the graph (b) pairwise but not higher order epistasis, and the graph (c) is compatible with additive fitness.

There are in total $1,862$ fitness graphs for the biallelic three-locus system, of which 698 graphs (37%) imply three-way epistasis. In principle one can check a particular three-locus system for higher order epistasis using this result. However, it is not convenient to work with a list of over one thousand graphs. In order to make the problem more tractable, we can utilize the fact that some fitness graphs are isomorphic (**Figure 5**). There are 54 distinct isomorphism classes of graphs for the three-

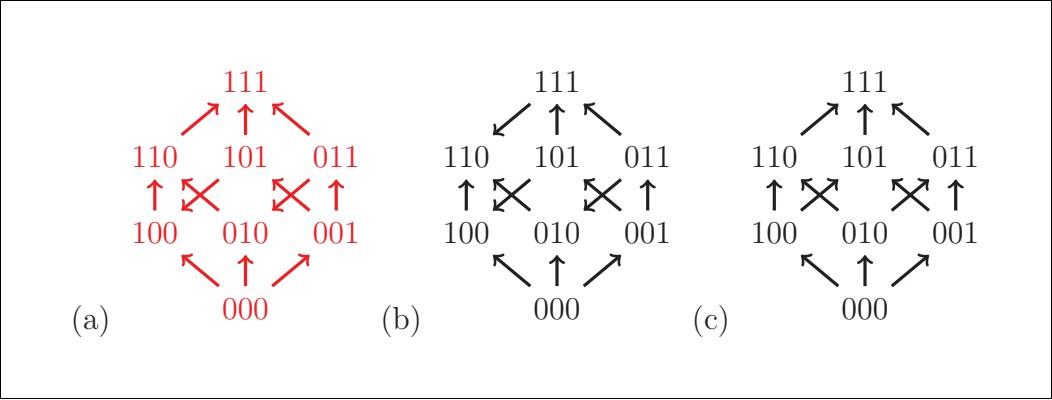

**Figure 4.** The fitness graph (**a**) implies three way epistasis, the graph (**b**) implies epistasis, but not higher order epistasis, and (**c**) does not imply epistasis, since all arrows point up.

DOI: https://doi.org/10.7554/eLife.28629.005

locus system, of which 20 imply higher order epistasis (Mathematical framework and proofs, Section Partial orders and fitness graphs). Consequently, to detect three-way epistasis, one can find the isomorphism class and then check 54 graphs, namely one for each isomorphism class (*Figure 6*).

We complete the consideration of the three-locus case by analyzing partial orders. Again, in favorable cases one can infer three-way epistasis. Indeed, if there exists a partition of all eight genotypes into four pairs $(g_e, g_o)$, where $e$ and $o$ are as above, and $w_{g_e} > w_{g_o}$ for each pair, then one can conclude three-way epistasis (Proposition 7).

## General $n$-locus case

The results on rank orders and higher order epistasis for $n = 3$ generalize to any number of loci. The definition of $n$-way epistasis in an $n$-locus system is analogous to the three-locus case, as is the condition for when rank orders imply $n$-way epistasis. Accordingly, a characterization of rank orders that imply $n$-way epistasis can be phrased in terms of Dyck words (Proposition 1). From this result it follows that the fraction of rank orders that imply $n$-way epistasis is $2/(2^{n-1} + 1)$ (Corollary 2) and that it can be determined in a computationally efficient manner whether or not a rank order implies $n$-way epistasis.

Rank order methods are useful for analyzing the total $n$-way epistasis for an $n$-locus system, as demonstrated. However, a single quantity cannot capture all possible gene interactions. Rank order approaches have the capacity to reveal finer interactions as well.

We start with a general description of gene interactions before exploring what rank orders can reveal about these interactions. We define an additive dependence relation as a linear form that is zero on additive fitness landscapes. Interaction coordinates and circuits (*Beerenwinkel et al., 2007*), as well as Walsh coefficients of order two or more (*Weinreich et al., 2013*), are additive dependence relations. For simplicity, we restrict our analysis to the three-locus system, although the arguments used are readily extendable to any number of loci $n$.

First, we consider additive dependence relations that directly correspond to the two-locus case by fixing one allele at the third locus. For example, if we fix the third locus at 0, then

$$a = w_{000} + w_{110} - w_{010} - w_{100}$$

measures pairwise epistasis between the first and second locus. Similarly, if we set the third locus to 1, then

$$b = w_{001} + w_{111} - w_{011} - w_{101}$$

measures pairwise epistasis between the first and second locus.

An example of an additive dependence relation with no correspondence in the two-locus setting is

$$m = w_{001} + w_{010} + w_{100} - w_{111} - 2w_{000},$$

which compares the fitness of the triple mutant 111 to the three single mutants. This expression is negative if the triple mutant has higher fitness than one would expect based on the fitness effects of the three single mutations. An example of an additive dependence relation with eight non-zero terms is

$$u_{110} = w_{000} - w_{100} - w_{010} + w_{001} + w_{110} - w_{101} - w_{011} + w_{111}.$$

One can verify that $u_{110}$ is twice the average of $a$ and $b$. For a systematic approach to a comprehensive set of gene interactions, one can take advantage of circuits, that is, minimal additive dependence relations. In contrast, $u_{110}$ is not minimal, because it can be derived from $a$ and $b$. There are 20 circuits for the three-locus system, including $a$, $b$, and $m$ (Mathematical framework and proofs, Section Circuits).

The interactions described by $a$ and $b$ are referred to as conditional epistasis, that is, interactions that measure the total epistasis of subsystems obtained by fixing some loci at 0 or 1. If the interactions $a$ and $b$ differ substantially, it may be important to consider both of them. However, if we are rather interested in the average interaction for the first two loci over all genetic backgrounds, then $u_{110}$ is the right measure.

In general, relations that measure average effects, such as $u_{110}$, are referred to as interaction coordinates. The interaction coordinates $u_{110}$ differs from the Walsh coefficient $E_{110}$ (*Weinreich et al., 2013*) only by a constant scaling factor. Provided that average effects are sufficient for the purpose of a study, one can analyze higher order epistasis by considering interaction coordinates, or Walsh coefficients, (*Weinreich et al., 2013*) (Mathematical framework and proofs).

One interesting class of circuits compares the effect of replacing pairs of loci with different backgrounds. For instance,

$$k = w_{000} - w_{001} - w_{110} + w_{111}$$

compares the effect of replacing 00 by 11 if the third coordinate is fixed at 0, versus if the third coordinate is fixed at 1. We refer to $k$ as a circuit measuring marginal epistasis between two pairs of loci as in *Beerenwinkel et al. (2007)*.

Arguments based on Dyck words can be used for analyzing rank orders and additive dependence relations in general. The letters are determined by the signs of the coefficients in the linear form, as for three-way epistasis (Theorem 3).

For each circuit and interaction coordinate, we identify all rank orders that determine its sign. The characterization is given in terms of general Dyck word conditions. We found that for each interaction coordinate, $2/5$ of all rank orders determine its sign; for each circuit corresponding to either conditional two-way interaction or marginal epistasis between two pairs of loci, $2/3$ of all rank orders determine its sign; and for each circuit relating the three-way interaction to the total two-way epistasis, $1/2$ of all rank orders determine its sign ( Corollary 4 and 6).

Importantly, if a rank order implies that the sign of an additive dependence relation, such as a circuit or an interaction coordinates, is determined, then the system has sign epistasis.

One can ask if it possible to decompose the word obtained for analyzing $n$-way epistasis into subunits, so as to learn about circuits or properties for subsystems of the genotypes. In general, no such decompositions are possible unless one has information in addition to the word itself. For instance, suppose that a rank order is mapped to *ooeeeooe*. The first half of the word, namely *ooee*, does not necessarily reveal any interesting information about the system. (Mathematical framework and proofs, Section Circuits).

The signs of all twenty circuits determines the polyhedral shape of the fitness landscape (*Beerenwinkel et al., 2007*). The shape combines the circuit information into a more manageable object. However, no rank order determines a shape for $n = 3$ (Mathematical framework and proofs, Section Circuits).

Our tools for detecting gene interactions work for total $n$-way epistasis, interaction coordinates and circuits. Moreover, our approach applies to any type of gene interaction that can be expressed by a linear form (Theorem 3), such as Fourier coefficients (*Beerenwinkel et al., 2007*) and Walsh coefficients (*Weinreich et al., 2013*). We have implemented algorithms for detecting the gene interactions described in this section, both for rank orders and partial orders, specifically, algorithms for $n$-way epistasis, three-way and four-way interaction coordinates, and three-way circuit interactions (https://github.com/gavruskin/fitlands#fitlands). The most computationally demanding task of our algorithms is reading the rank order from disk. The run time complexity is proportional to the number of non-zero coefficients in the linear form.

## Analysis of empirical fitness data

As proof of principle, we applied our tools to fitness data from a diverse set of biological systems, ranging from HIV-1 (*Segal et al., 2004*), malaria (*Ogbunugafor and Hartl, 2016*), antibiotic resistance (*Mira et al., 2015*), to the fungus *Aspergillus Niger* (*Franke et al., 2011*). Our approach reveals higher order epistasis for all of these systems, only by considering rank orders and partial orders of genotypes, without the need to access direct fitness measurements or estimates.

Our first application is to the HIV-1 data published by *Segal et al., 2004*. Following *Beerenwinkel et al. (2007)*, we consider the three-locus biallelic system that consists of the mutation L90M in the protease and mutations $\mathrm{M184V}$ and $\mathrm{T215Y}$ in the reverse transcriptase of HIV-1. Fitness was measured as the number of offspring in a single replication cycle of the virus in the original study, and was reported relative to the wild-type strain NL4-3 on a logarithmic scale. The data consist of 288 fitness measurements, including between 5 and 214 replicates per genotype.

The following rank order was obtained by comparing the mean fitness of the eight genotypes:

$$w_{000} > w_{100} > w_{011} > w_{110} > w_{101} > w_{001} > w_{010} > w_{111},$$

where 000 corresponds to the sequence of amino acids $\mathrm{LMT}$ and 111 to $\mathrm{MVY}$ comprising the three selected loci. This rank order implies positive three-way epistasis because the associated word

$$eoeeeooo$$

is a Dyck word. It follows that the three mutations under consideration together have a stronger effect on fitness than one would predict from single and double mutants. A closer inspection of the word reveals more information. If we swap any two adjacent letters in the word, then we still have a Dyck word, with the single exception of the first two letters. In other words, only one pair of adjacent genotypes in the rank order, namely 000 and 100, could violate the conclusion if transposed.

If the experiment was to continue, our analysis could be used to direct the data collection process. Indeed, the argument above suggests that the position of the genotype 100 may violate the conclusion of positive three-way epistasis. To quantify the uncertainty in the ranking of 100 with respect to the wild type 000, we employed the Wilcoxon rank sum test on the replicate fitness measurements. The p-value of the test is 0.47 for the relation $w_{000} > w_{100}$, which implies considerable uncertainty and justifies our recommendation of further experiments to clarify the position of 100. Importantly, the suggested experiment reduces the number of measurements required to make a more robust conclusion about epistasis considerably, namely to one out of 28 possible comparisons.

We proceeded by considering other types of gene interactions in this data set. When considering all 20 circuits, the rank order implies interactions for 55% of the circuits, with positive sign for 30% and negative for 25% of the circuits. This result is consistent with the conventional statistical approaches that use direct fitness measurements. Indeed, since the empirical study of *Segal et al., 2004* provided multiple fitness measurements of each genotype, it was possible to compare our conclusion based on rank orders with statistical tests based on fitness estimates.

The results for fitness measurements were confirmed by the conventional Wilcoxon rank sum test. We computed interaction coordinates and circuit interactions for the summary statistics reported in (*Beerenwinkel et al., 2007*). *Figure 7* shows that our rank order methods detected the majority of circuit interactions identified by using the summary statistics. Specifically, both approaches detected three-way epistasis. Furthermore, 11 of the 20 circuit interactions have been detected by our method and confirmed by Wilcoxon rank sum test.

We also applied Student's t-test to detect interactions and quantify the significance of the estimates. In addition to the 11 interactions, the t-test found 6 circuit interactions as significant (Table 2). We emphasize that the rank order approach required much less information to arrive at the same conclusions, thus demonstrating the power of the method.

We conclude that the three sites in the HIV-1 genome under consideration are prone to a diverse set of interactions. Specifically, the strong support for the three-way epistasis, along with the 55% of informative circuit interactions, imply that the three loci together interact in a complex manner, meaning that the interactions cannot be explained using pairwise interactions alone. Thus, in this data set, higher order interactions have a strong impact on the fitness landscape.

Our second application is to a study of antimicrobial drug resistance in malaria (*Ogbunugafor and Hartl, 2016*). The authors measured growth rates for several mutants of *Plasmodium vivax* under exposure to the antimalarial drug pyrimethamine. We identified higher order epistasis by analyzing rank orders. More precisely, we considered a three-locus sub-system of the study that consists of mutations N50I, S58R, and S117N, in the context of T173L, a fixed mutation, under nine different concentrations of pyrimethamine. The genotypes comprising positions 50, 58, and 117 are labeled 000 (NSS), 100, 010, 001, 110, 101, 011, and 111 (IRN). The three highest concentrations of the drug resulted in the following rank orders:

$$w_{111} > w_{011} > w_{001} > w_{101} > w_{010} > w_{100} > w_{110} > w_{000}$$
$$w_{111} > w_{011} > w_{001} > w_{010} > w_{100} > w_{101} > w_{110} > w_{000}$$
$$w_{111} > w_{011} > w_{010} > w_{001} > w_{100} > w_{110} > w_{101} > w_{000}.$$

The corresponding words are *oeoeooee* for the first rank order, obtained under the highest

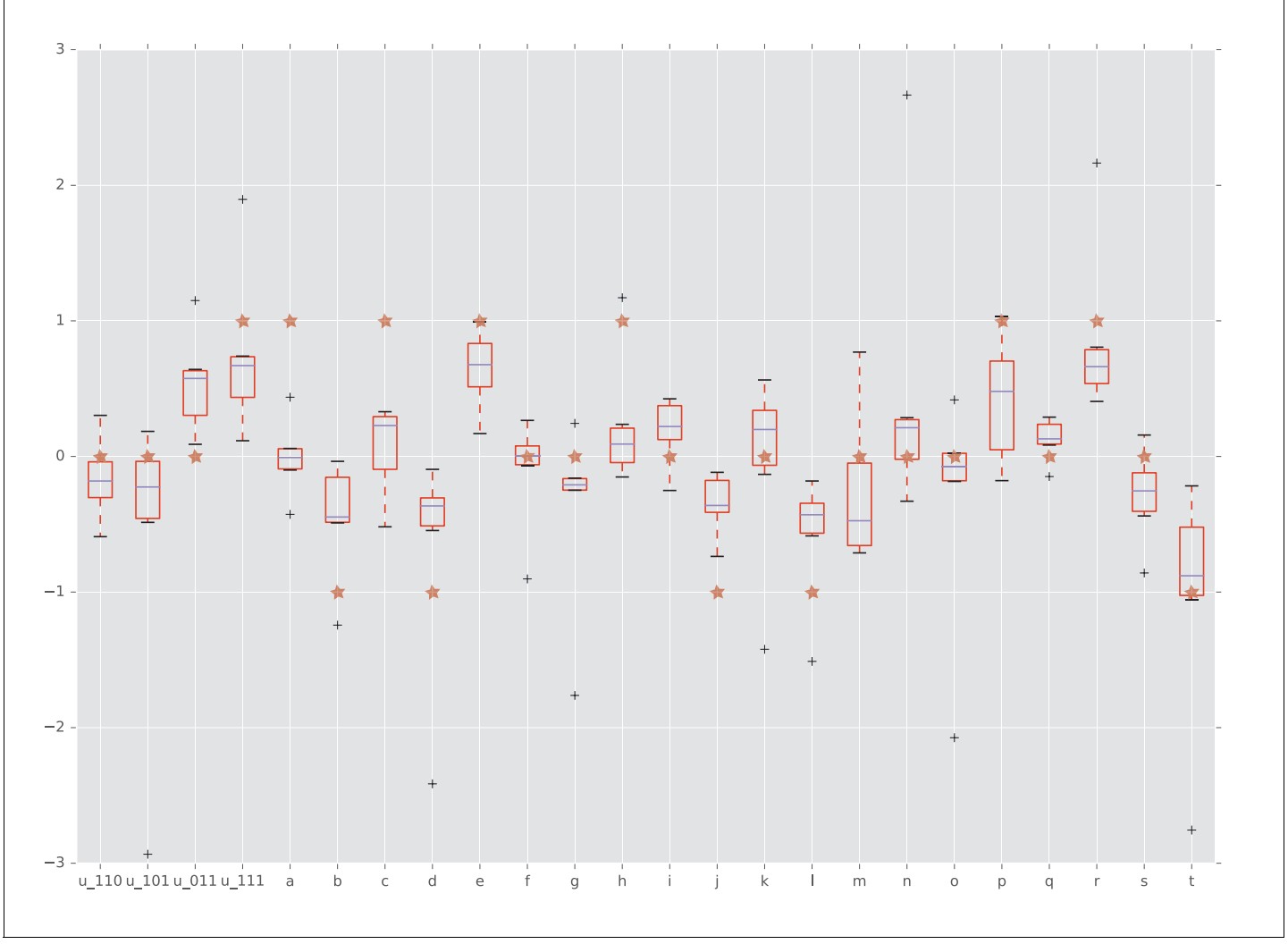

**Figure 5.** Interactions detected from fitness summary statistics and from rank orders. The horizontal axis is labeled by the four interaction coordinates $u_{110}, \ldots, u_{111}$ and twenty circuits $a, \ldots, t$. The boxplots show the distributions of the various interactions induced by the empirical fitness distribution. The red star indicates whether the interaction has been detected by our rank order method. Specifically, a star with vertical coordinate $-1$, 0, and one means negative, no, and positive interaction, respectively.

DOI: https://doi.org/10.7554/eLife.28629.006

concentration of the drug, and *oeoooeee* for the second and third rank orders. Since we obtain Dyck words in all cases, the system has negative three-way epistasis for the three highest concentrations of the drug. This consistency among pyrimethamine concentrations shows that the result is robust.

Using our software, we also analyzed all interaction coordinates for this data set (https://github.com/gavruskin/fitlands/blob/master/Four-way_interaction_coordinates_and_total_n-way_interaction.ipynb). Our analysis revealed that for the two highest concentrations of the drug, the rank order implies that the interaction coordinate denoted $u_{0111}$(Mathematical framework and proofs) is negative.

Next, we applied our tools to a study of the TEM-family of $\beta$-lactamase, associated with antibiotic resistance (*Mira et al., 2015*). The study measured growth rates for 16 genotypes exposed to 15 different antibiotics. Specifically, all 16 genotypes that combine subsets of the four amino acid substitutions M69L, E104K, G238S, N276D found in TEM-50, including eight known enzymes, were created using site-directed mutagenesis. We considered the fitness graph obtained when the system was exposed to the antibiotic FEP Cefepime at a concentration of 0.0156 $\mu$g/ml (*Figure 8*). The fitness graph implies higher order epistasis (Proposition 7), that is, the fitness of TEM-50 cannot be

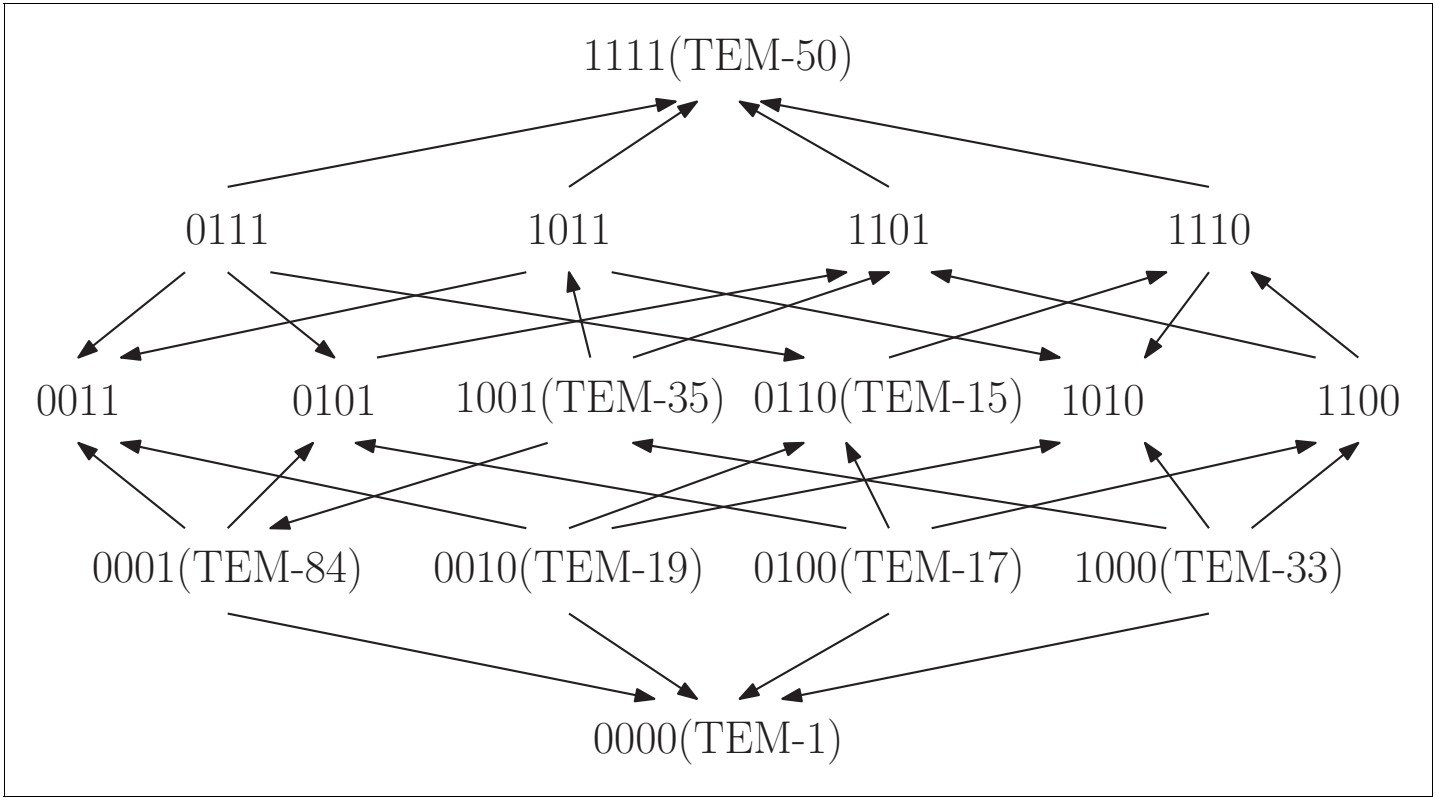

**Figure 6.** The TEM-family of *β*-lactamase contributes to antibiotic resistance problems in hospitals. The fitness graph shows a four-locus system consisting of the wild type, TEM-1, the quadruple mutant, TEM-50, and all intermediate mutants, including six clinically found mutants in the TEM family. The mutation M69L corresponds to 1000, E104K to 0100, G238S to 0010, and N276D to 0001. Growth rates were measured for the 16 genotypes under exposure to the antibiotic FEP Cefepime, and the fitness graph was determined accordingly (*Mira et al., 2015*). The graph reveals higher order epistasis.

DOI: https://doi.org/10.7554/eLife.28629.007

predicted even with complete knowledge of the fitness values of the remaining genotypes in the system. The fitness graphs for the other 14 antibiotics do not share this property. We conclude that even though some of the single and triple mutants confer low antibiotic resistance, a large population of triple mutants alone is more prone to become antibiotic resistant due to the epistatic fitness advantage of TEM-50, as compared to a setting with no higher order epistasis.

Finally, we investigated a study of the filamentous fungus *Aspergillus Niger* (*Franke et al., 2011*). We considered a system consisting of the wild type and all combinations of the four individually deleterious mutations *fwnA1*, *leuA1*, *oliC2* and *crnB12* (*Figure 9*). Fitness was estimated with two-fold replication by measuring the linear mycelium growth rate in the original study. The fitness graph implies higher order epistasis (Proposition 7).

All four arrows incident to 0000 point towards the genotype, so that the genotype 0000 is a peak in the landscape. The same is true for the genotypes 1100, 0011, and 1001. Because of the four peaks, it is possible that the fungus population gets stranded at a suboptimal peak during the course of evolution (we do not necessarily assume that the starting point for an evolutionary process is at 0000). In contrast, an additive fitness landscape is single peaked. This example illustrates that epistasis may have an impact on the evolutionary dynamics. Several peaks can make the evolutionary process less predictable, depending also on other factors such as population size, mutation rate, etc. More generally, for three-locus fitness graphs, we analyzed the impact of higher order epistasis versus only pairwise epistasis systematically. We found that higher order epistasis correlates with more peaks as well as other features that can lead to involved evolutionary dynamics (Mathematical framework and proofs, Section Graph theoretical aspects).

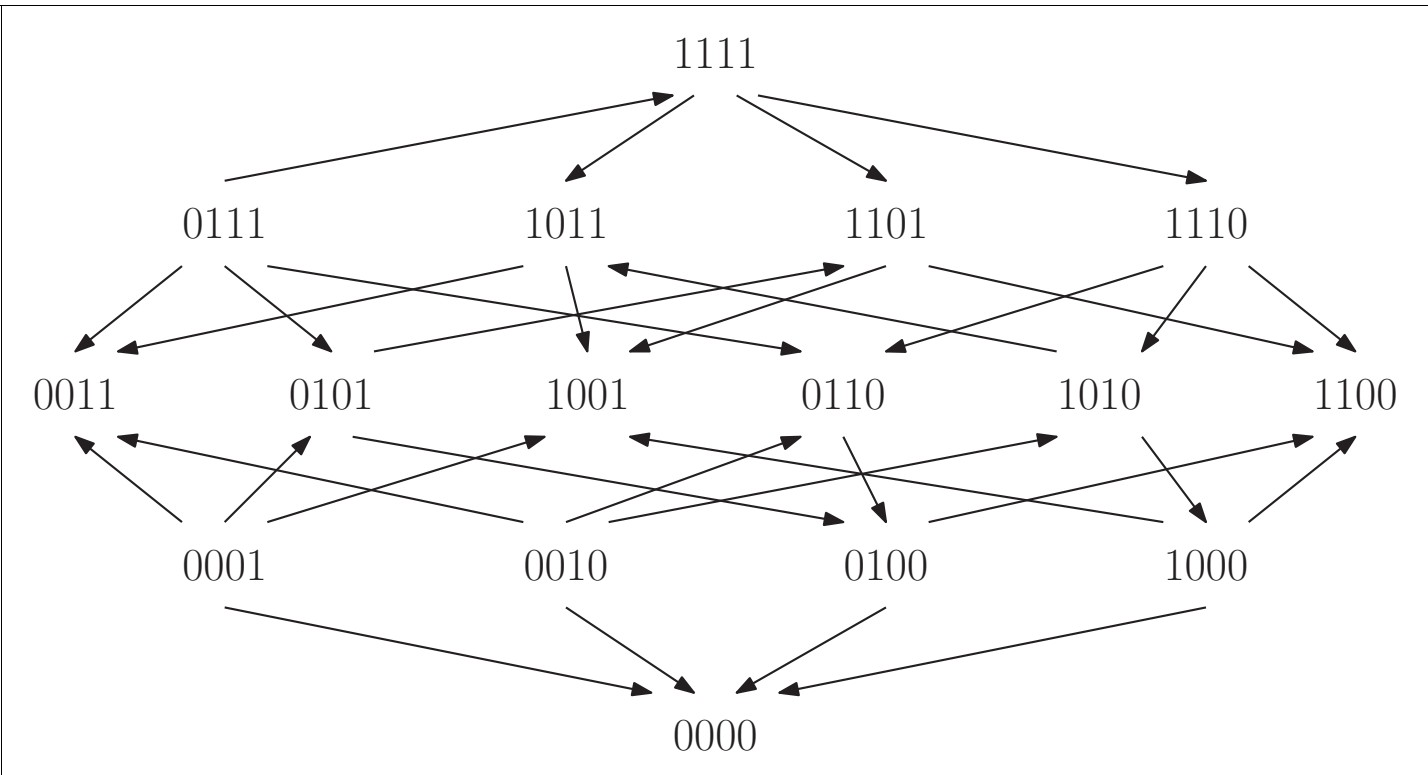

**Figure 7.** The fitness graph shows a four-locus system for the filamentous fungus *Aspergillus Niger*. The system consists of all combinations of the four individually deleteriouis mutations *fwnA1*, *leuA1*, *oliC2* and *crnB12*. The landscape has in total four peaks, labeled 0000, 1100, 0011 and 1001.
DOI: https://doi.org/10.7554/eLife.28629.008

We completed our analysis of this data set by considering the 5-locus system of mutations *fwnA1*, *argH12*, *pyrA5*, *leuA1*, and *pheA1*, conditioning on mutations *lysD25*, *oliC2*, and *crnB12* all being absent. The original study does not contain any measurements for the two genotypes 11010 and 10111. Furthermore, two pairs of genotypes in the system have identical ranks, namely (11000, 10010) and (10011, 11101). We obtained the following word for the total five-way epistasis $u_{11111}$,

$$eeoeexxooooeoooeoyyeeeoeoeoeoo$$

where the two $x$'s correspond to genotypes 11000 and 10010, the $y$'s to 10011 and 11101, and two letters are missing. Whether one can draw conclusions about five-way epistasis or not, depends on the positions of the two missing genotypes, as well as the genotypes represented by $y$'s, whereas it is independent of the genotypes represented by $x$'s. Specifically, if genotype 11101 has higher fitness than 10011, genotype 10111 has rank between 1 and 15, and genotype 11010 has rank between 20 and 32, then the resulting rank order implies positive five-way epistasis, that is, $u_{11111}>0$, for both possible options to resolve the ranking of genotypes 11000 and 10010 (*Table 3*).

## Discussion

Gene interactions play a critical role in evolutionary processes. Important features of fitness landscapes, such as the number of peaks, and accessible evolutionary trajectories, depend on epistatic gene interactions. The importance of higher order versus pairwise epistasis, within and among genes or in non-coding regions, as well as the impact of higher order epistasis on evolutionary dynamics, remains a central research topic (*Sailer and Harms, 2017b*; *Wu et al., 2016*; *Weinreich et al., 2013*). Progress in all of these areas requires adequate mathematical and statistical approaches, in addition to empirical studies.

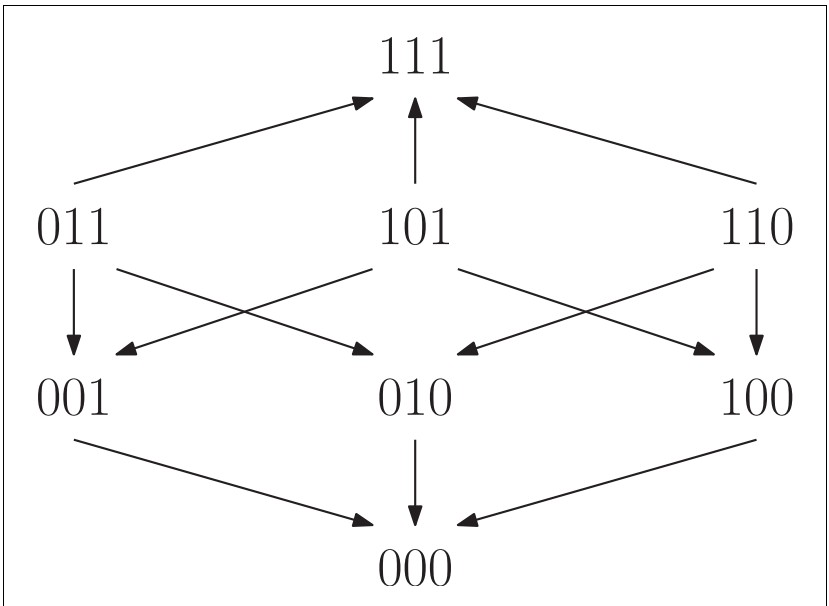

**Figure 8.** The fitness graph is compatible with the two rank orders (5) and (6).
DOI: https://doi.org/10.7554/eLife.28629.009

Here, we have developed new quantitative tools for detecting gene interactions from empirical data. The main advantage of our tools is that they can reveal gene interactions from the types of data frequently generated in empirical studies, specifically rank orders, fitness graphs, and general partial orders of genotypes. The reasons why, in practice, these types of data are available more often than precise fitness measurements for each genotype are manifold. They include restricted comparative experimental designs and known and unknown confounding factors in measuring fitness that can result in uncertain and biased estimates. The methods presented here allow for studying epistatic interactions even when direct fitness measurements are lacking or only a subset of pairwise fitness comparisons is available, either as the immediate outcome of the experiment or the reported summary.

We provide a complete characterization of rank orders that imply higher order epistasis, along with precise results for fitness graphs of three-locus systems. In principle, our approach applies to general partial orders as well, and we have implemented algorithms accordingly. However, because of the increasing computational complexity it would be desirable to have theoretical results for handling large systems. In particular, a characterization of fitness graphs that imply higher order epistasis is of independent mathematical interest.

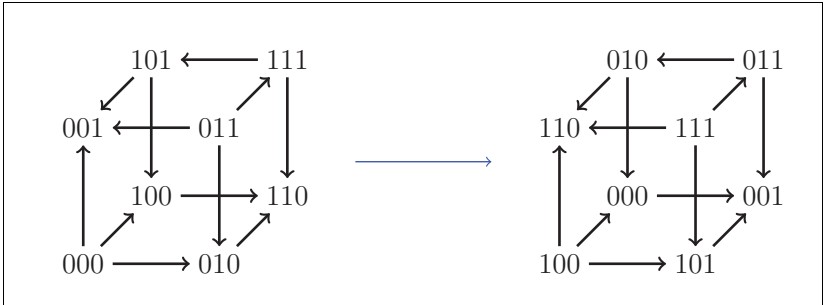

**Figure 9.** An example of an isomorphism. Here, the allele labels '0' and '1' in the first locus have been interchanged, as well as the second and third loci.
DOI: https://doi.org/10.7554/eLife.28629.010

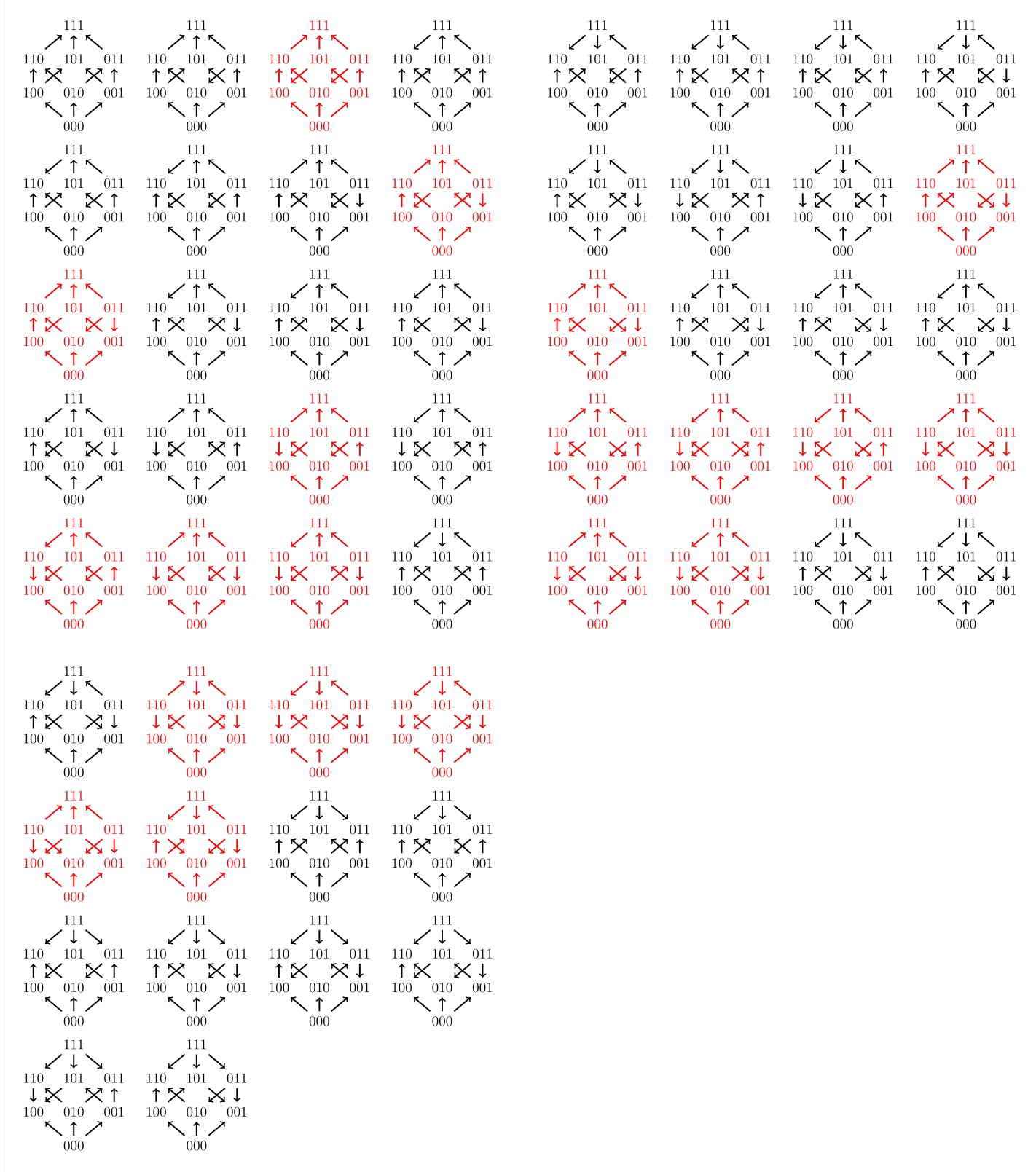

**Figure 10.** All 54 fitness graph types. Those depicted in red imply three-way epistasis.

DOI: https://doi.org/10.7554/eLife.28629.012

We found that for biallelic three-locus systems, 40% of all possible rank orders and 37% of all possible fitness graphs imply higher order epistasis. These fractions suggest that our methods have a good capacity to detect higher order epistasis among three loci, even if exact and complete fitness measurements or estimates are not available.

The fraction of rank orders that imply $n$-way epistasis decreases rapidly with increasing number of loci, $n$. However, many rank orders are informative regarding other additive dependence relations, for instance some circuits measuring conditional epistasis. This is clear from the observation that the proportion of rank orders that are compatible with additive fitness decreases rapidly with $n$.

Moreover, the power of our methods was demonstrated for a diverse set of biological systems. We detected higher order epistasis for HIV, malaria, the fungus *Aspergillus Niger*, and antibiotic resistance systems. Our findings suggest that genetic interactions beyond two-way epistasis shape the fitness landscapes of these genetic systems and may play an important role in determining their evolutionary trajectories. We also exhaustively investigated various types of higher order interactions in HIV-1 and discovered a complex pattern of interactions, confirming that our approach is powerful enough to detect finer gene interactions. Specifically, we identified over twenty interactions by conventional approaches, and rank order methods detected about half of them.

Another important application of our method is to experimental design. When the information available in the data does not contradict an interaction, but is not conclusive enough to claim the interaction, for example because the number of performed competition experiments is too small, then the method allows for prioritizing further experiments by suggesting additional comparisons of genotypes. This feature may prove useful in guiding fitness experiments that aim for testing specific interactions and allow for iteration. We have developed this idea further in *Lienkaemper et al. (2017)*, where we consider partial fitness orders of genotypes and develop efficient algorithms to detect genetic interactions, as well as study the geometry of such partial orders. Evolutionary aspects of partial orders and gene interactions are studied in *Crona and Luo, 2017*.

Genetic interactions, especially those of higher order, are particularly difficult to detect in high-dimensional systems, where complete fitness measurements of all genotypes are infeasible. Human genome-wide association studies (GWAS) are a prime example. Here, already the number of loci, $n$, and certainly the number of possible genotypes, $2^n$, is much larger than the actual number of genotyped and phenotyped individuals, even if the genotype data is summarized on the level of genes or haplotype blocks. Since most diseases are polygenic, rather than monogenic, genetic interactions play an important role, and accounting for them may explain some of the missing heritability and improve genetic disease models. Several methods have been proposed for detecting pairwise interactions in GWAS, most of them relying on scanning all or a prioritized subset of pairs of loci (*Wei et al., 2014*), but little is known about higher order interactions in these landscapes.

The methods presented in the present study may help addressing this challenge as they can sometimes reveal higher order interactions from a small number of comparisons, and the choice of genotypes to compare can be optimized if particular interactions are to be tested. Another advantage is the flexibility of our approach regarding the type of epistatic interaction analyzed. While we have focused on analyzing complete genetic systems, that is, $n$-dimensional hypercubes, for small values of $n$ in this work, genotype spaces consisting of subsets of the $2^n$ possible genotypes have different sets of interactions, such as circuits, that are natural to consider. Towards this end, *Huggins et al. (2007)* have explored circuits and their sign patterns for genotype data from the HapMap project in two ENCODE regions. The Dyck word approach will be particularly useful if quantitative data on the phenotype is difficult to obtain, but rank order information is more accessible, for example by considering disease indicators, rather than the condition itself.

For a more theoretical perspective, we emphasize the distinction between rank order-induced gene interactions, and interactions that do not change the rank order of genotypes. This distinction was pointed out by *Weinreich et al., 2005* who introduced the term sign epistasis. If a system has sign epistasis, then the rank order of the genotypes is not compatible with additive fitness. Rank order-induced gene interactions of any type can thus be regarded as analogues to sign epistasis.

There exist a number of possible ways to quantify and interpret higher order interactions (*Weinreich et al., 2013*; *Beerenwinkel et al., 2007*; *Hallgrímsdóttir and Yuster, 2008*), and our rank order approach applies to any type of gene interactions measured by linear forms. In particular, we can detect interactions as described by Fourier coefficients and Walsh coefficients. From our general argument based on Dyck words we investigated three-locus systems, and determined the

number of rank orders that imply circuit interactions, including conditional and marginal epistasis, and similarly for interactions coordinates. The method works equally well for other interactions.

Further investigation of rank order-induced interaction has the potential to relate global and local properties of fitness landscapes, similarly to results on sign epistasis (*Weinreich et al., 2005*; *Poelwijk et al., 2011*; *Crona et al., 2013*). Global properties concern peaks and mutational trajectories in the fitness landscape, whereas local properties concern, for instance, fitness graphs for small system. The relation between global and local properties is important since only local properties can be easily observed in experiments or nature.

A very useful feature of sign epistasis is that one can identify, or rule out, sign epistasis in a system from inspection of two-locus subsystems only. The theoretical significance and applicability of sign epistasis depends on its local nature. Fortunately, the signed versions of other gene interactions are sometimes local as well. For instance, it seems plausible that the absence of rank order induced conditional epistasis of specified orders (a local property) correlates with few peaks and good peak accessibility. In this spirit we explore some evolutionary consequences of higher order epistasis in *Crona and Luo, 2017*. Theory for sign epistasis stands as a model for further development in this area.

Although we have applied our method here only to fitness, any other continuous phenotype of interest can be analyzed in exactly the same manner. The fitness landscape $w$ is then replaced by a more general genotype-to-phenotype map. For example, rather than using it as a fitness proxy, one may be concerned about the drug resistance phenotype itself and its genetic architecture.

In summary, rank order methods have potential for the interpretation of empirical data, as well as for relating higher order gene interactions and evolutionary dynamics. Our approach facilitates detecting higher order epistasis from a very broad range of empirical data, and will therefore contribute to enhancing our general understanding of empirical fitness landscapes and epistatic gene interactions.

## Materials and methods

### Mathematical framework and proofs

Here we provide proofs for the results in the main text, and give a brief background on the discrete Fourier transform, Dyck words, and Catalan numbers. Catalan numbers (*Stanley, 1999*) have rarely been used in biology, so we describe them briefly without assuming any knowledge. Several arguments in the Materials and methods depend on elementary combinatorics, and the reader may consult a general text, such as *Grimaldi (2006)*.

We start with rank orders and the total $n$-way epistasis, followed by more general results on rank orders, circuits and other linear forms. The next topic is epistasis and fitness graphs, including some related graph theory. Finally we provide a few observations on epistasis and partial orders.

Gene interactions for a biallelic $n$-locus system can be described in terms of the Fourier transform for $(\mathbb{Z}_2)^n$ defined as

$$u_{i_1 i_2 \ldots i_n} = \frac{1}{2^{n-1}} \cdot \sum_{j_1=0}^{1} \sum_{j_2=0}^{1} \cdots \sum_{j_n=0}^{1} (-1)^{i_1 j_1 + i_2 j_2 + \ldots + i_n j_n} w_{j_1 j_2 \ldots j_n}.$$

By abuse of notation we will ignore the scaling factor $\frac{1}{2^{n-1}}$. We define interaction coordinates as the elements $u_{i_1 i_2 \ldots i_n}$ such that at least two entries in $i_1 i_2 \ldots i_n$ are 1. The interaction coordinate $u_{1\ldots1}$ measures the total $n$-way epistasis,

$$u_{1\ldots1} = \sum_{j_1=0}^{1} \sum_{j_2=0}^{1} \cdots \sum_{j_n=0}^{1} (-1)^{j_1 + j_2 + \ldots + j_n} w_{j_1 j_2 \ldots j_n}.$$

In particular,

$$u_{11} = w_{00} - w_{10} - w_{01} + w_{11} = \varepsilon_2$$

and

**Table 1.** Numbers and fractions of rank orders that imply $n$-way epistasis.

| Loci | Rank orders | Imply epistasis | Fraction |
| --- | --- | --- | --- |
| 2 | 24 | 16 | 2/3 |
| 3 | 40,320 | 16,128 | 2/5 |
| 4 | 20,922,789,890,000 | 4,649,508,864,000 | 2/9 |

DOI: https://doi.org/10.7554/eLife.28629.011

$$u_{111} = w_{000} - w_{100} - w_{010} - w_{001} + w_{110} + w_{101} + w_{011} - w_{111} = \varepsilon_3$$

as defined in *Equations 1 and 2* in the main text. A biallelic three-locus system has three-way epistasis exactly if $u_{111} \neq 0$. Otherwise the system has only pairwise interactions. Similarly, a biallelic $n$-locus system has (total) $n$-way epistasis exactly if $u_{1\ldots1} \neq 0$.

The remaining interaction coordinates of the three-locus system are

$$u_{110} = w_{000} - w_{100} - w_{010} + w_{001} + w_{110} - w_{101} - w_{011} + w_{111},$$
$$u_{101} = w_{000} - w_{100} + w_{010} - w_{001} - w_{110} + w_{101} - w_{011} + w_{111},$$
$$u_{011} = w_{000} + w_{100} - w_{010} - w_{001} - w_{110} - w_{101} + w_{011} + w_{111}.$$

*Weinreich et al. (2013)* characterize epistatic interactions using Walsh coefficients, which are closely related to interaction coordinates. Specifically, the Walsh coefficients $E_{111}$, $E_{110}$, $E_{101}$, $E_{011}$ differ from the interaction coordinates $u_{111}$, $u_{110}$, $u_{101}$, $u_{011}$ only by a scalar. Here we ignore scalars, since we focus on the signs of the interactions coordinates only. It follows that all our results on interaction coordinates hold for Walsh coefficients as well.

## Rank orders

We will determine the number of rank orders which imply $n$-way epistasis. The proof depends on Catalan numbers and Dyck words (*Stanley, 1999*). Let $C_i$ denote the $i$th Catalan number for $i \geq 0$, that is, $C_i = \frac{(2i)!}{(i+1)!i!}$. In particular, $C_0 = C_1 = 1, C_2 = 5, C_3 = 14$ and $C_4 = 42$. A Dyck word of length $2n$ in the letters $X$ and $Y$ is a string consisting of $n$ $X$'s and $n$ $Y$'s such that no initial segment of the string has more $Y$'s than $X$'s. For instance, the Dyck words of length 4 are $XXYY$ and $XYXY$. The initial segments of $XXYY$ are $X$, $XX$, $XXY$, and $XXYY$.

**Proposition 1.** *Consider a biallelic n-locus system. The number of rank orders which imply n-way epistasis is*

$$\frac{(2^n)! \times 2}{2^{n-1} + 1}$$

*Proof.* There are $(2^n)!$ rank orders in total. Let $e_i$ denote the fitnesses of genotypes with an even number of 1's in the subscripts ($w_{0\ldots0}$, $w_{110\ldots0}$, and so forth) and $o_i$ the fitnesses of genotypes with an odd number of 1's, ordered in such a way that $e_i > e_{i+1}$ and $o_i > o_{i+1}$ for all $i$. We will refer to even and odd elements from now on. Let $u_{1\ldots1}$ denote the interaction coordinate as defined above. Notice that $u_{1\ldots1} = 0$ exactly if $\sum_i e_i - \sum_i o_i = 0$. Consequently a rank order implies positive $n$-way epistasis ($u_{1\ldots1} > 0$) when the sum $\sum_i (e_i - o_i)$ is positive for all fitness landscapes compatible with the rank order. It is therefore sufficient to count such rank orders.

We define a map from fitness rank orders to words in the alphabet $\{e, o\}$ as follows: $e_i \mapsto e, o_i \mapsto o$. For instance, the order $w_{00} > w_{11} > w_{10} > w_{01}$ is mapped to $eeoo$. We claim that a rank order satisfies $\sum_i (e_i - o_i) > 0$ exactly when it is mapped to a Dyck word (where $e$ precedes $o$).

It is immediate that $\sum_i (e_i - o_i) > 0$ holds if the rank order is mapped to a Dyck word. Conversely, suppose that a rank order is not mapped to a Dyck word. Let $s$ be the least number such that the number of $o$'s exceeds the number of $e$'s for an initial segment of length $s$ (note that $s$ has to be odd in this case) and let $j = \frac{s+1}{2}$. Clearly one can make the sum of $\sum_{i=1}^{j} (e_i - o_i)$ negative for a particular choice of $e_i$ and $o_i$. By choosing the remaining numbers $e_i, o_i$ sufficiently small, we get $\sum_i (e_i - o_i) < 0$, which proves the claim.

It remains to count the rank orders where $\sum_i (e_i - o_i) > 0$. Such rank orders are mapped to Dyck words (where $e$ precedes $o$) consisting of $2^{n-1}$ $X$'s and $2^{n-1}$ $Y$'s. There are $C_{2^{n-1}}$ such Dyck words (**Stanley, 1999**). For each word there are $(2^{n-1})! \times (2^{n-1})!$ fitness rank orders which map to the word. Indeed, one can choose the ordering of even and odd elements each in $(2^{n-1})!$ different ways.

In total there are $C_{2^{n-1}} \times (2^{n-1})! \times (2^{n-1})!$ fitness rank orders such that $\sum_i (e_i - o_i) > 0$ for all landscapes. By symmetry, the same number of fitness rank orders satisfy the negative epistasis condition $\sum_i (e_i - o_i) < 0$. One verifies that

$$C_{2^{n-1}} \times (2^{n-1})! \times (2^{n-1})! \times 2 = \frac{(2^n)! \times 2}{(2^{n-1} + 1)},$$

which completes the proof. □

A few observations in the proof of Proposition 1 are of interest. Importantly, the proof gives a computationally efficient method (linear in the number of genotypes) for checking if a rank order implies $n$-way epistasis. Indeed, the rank order implies higher order epistasis exactly if it is mapped to a Dyck word. Moreover, the proposition states that $\frac{(2^n)! \times 2}{(2^{n-1} + 1)}$ orders imply $n$-way epistasis. From the proof it is clear that half of these orders imply positive $n$-way epistasis ($u_{1\dots1} > 0$) and the other half negative $n$-way epistasis ($u_{1\dots1} < 0$). Also, the proof points out some symmetries. If a rank order implies epistasis, then the same is true for rank orders obtained by (i) any permutation of the even elements, (ii) any permutation of the odd elements, and (iii) the flip obtained by replacing every "<" by ">" in the rank order. It follows that each rank order that implies three-way epistasis belongs to a class of 1152 elements, which differ by the operations (i)–(iii) only.

**Corollary 2.** *The fraction of rank orders that imply $n$-way epistasis among all rank orders is*

$$\frac{2}{2^{n-1} + 1}.$$

*Proof.* Since the number of all rank orders is $(2^n)!$, the result follows. □

The results on rank orders and epistasis for $2 \leq n \leq 4$ are summarized in **Table 1**. Notice that the expression in the corollary approaches $\frac{1}{2^{n-2}}$ for large $n$.

Interestingly, no integer sequence that starts with 16, 16 128, . . . is available at **The On-Line Encyclopedia of Integer Sequences (2016)**.

## Circuits

The proof of Proposition 1 depends on the map defined from the rank orders to words in the alphabet $\{e, o\}$. We will use a generalization of the map in subsequent proofs. The starting point is a given linear form. The form determines a map from rank orders to words. Although the idea is closely related to the previous proof, we will work with positive and negative coefficients in the linear forms. For that reason, we will use $P$ and $N$ rather than $e$ and $o$ (even and odd is no longer meaningful).

We start with a clarifying example. Assume that a given linear form has integer coefficients and that the sum of its coefficients is zero. For instance, the form

$$m = w_{001} + w_{010} + w_{100} - w_{111} - 2w_{000}$$

defines a map $\varphi^m$ as follows: Each of the variables $w_{001}, w_{010}, w_{100}$ corresponds to the letter $P$ (for positive), and the variable $w_{111}$ corresponds to $N$ (for negative). The variable $w_{000}$ corresponds to $NN$, because of the coefficient $-2$. In this case, the rank order

$$w_{111} > w_{001} > w_{000} > w_{100} > w_{010} > w_{110} > w_{101} > w_{011}$$

is mapped to $NPNNPP$ under $\varphi^m$. Specifically, starting from left $w_{111}$ corresponds to $N$, $w_{001}$ to $P$, $w_{000}$ to $NN$, $w_{100}$ to $P$, and $w_{100}$ to $P$. The remaining variables $w_{110}, w_{101}, w_{011}$ do not impact the word, since their coefficients are zero for the form $m$.

**Definition 1.** Let $f$ be a linear form with integer coefficients. Assume that the sum of its coefficients is zero. Let $\varphi^f$ denote the map from a total order on the variables (a rank order) to words in the alphabet $\{P, N\}$ defined as follows: Each variable of $f$ with a positive integer coefficient $c$ corresponds to a substring of $c$ letters $P$. Each variable in $f$ with a negative integer coefficient $c'$ corresponds to a substring of $|c'|$ letters $N$. A rank order of the variables is mapped to the word

consisting of the substrings obtained for each variable with non-zero coefficient in $f$. Specifically, the substrings (from left to right) of the word correspond to the variables in the rank order (from highest to lowest fitness).

The proof of Proposition 1 uses $e$ and $o$ instead of $P$ and $N$. However, notice that the map from rank orders to words in $e$'s and $o$'s is exactly $\varphi^{u_{1\ldots1}}$ (modulo the labeling). The next result is a generalization of Proposition 1. The proof is similar in every step with the modification that $\varphi^{u_{1\ldots1}}$ is replaced by $\varphi^f$ for an arbitrary form $f$, so we omit the details.

**Theorem 3.** *Let $f$ be a linear form with integer coefficients. Assume that the sum of its coefficients is zero. Then a rank order implies that $f$ is not zero if and only if it is mapped to a Dyck word by $\varphi^f$.*

We define additive dependence relations as linear forms that are zero for all additive landscapes. Theorem 3 applies to all additive dependence relations, because the coefficients of an additive dependence relation sum to zero. This fact explains why our Dyck word-based method applies broadly. All we require is that an epistasis measure is defined by linear forms, as any such measure is zero on additive fitness landscapes. In particular, Theorem 1 applies to $n$-way epistasis, interaction coordinates, and circuits. However, some approaches to epistasis are of a completely different type, for instance the approach based on Shannon entropy (*Moore et al., 2006*), in which case rank order methods may not apply.

Recall from the main text that $a = w_{000} - w_{010} - w_{100} + w_{110}$ is a circuit. In particular, $a = 0$ for all additive fitness landscapes. Moreover, $a$ is minimal with this property, in the sense that no linear form in a proper subset of $\{w_{000}, w_{010}, w_{100}, w_{110}\}$ equals zero for all additive landscapes. In general, circuits are defined as minimal (additive) dependence relations, in the sense that the set of $w_g$ which appear with non-zero coefficient is minimal with respect to inclusion.

There are 20 circuits $a, \ldots, t$ for the three-locus system (*Beerenwinkel et al., 2007*), namely

$$a := w_{000} - w_{010} - w_{100} + w_{110}$$
$$b := w_{001} - w_{011} - w_{101} + w_{111}$$
$$c := w_{000} - w_{001} - w_{100} + w_{101}$$
$$d := w_{010} - w_{011} - w_{110} + w_{111}$$
$$e := w_{000} - w_{001} - w_{010} + w_{011}$$
$$f := w_{100} - w_{101} - w_{110} + w_{111}$$
$$g := w_{000} - w_{011} - w_{100} + w_{111}$$
$$h := w_{001} - w_{010} - w_{101} + w_{110}$$
$$i := w_{000} - w_{010} - w_{101} + w_{111}$$
$$j := w_{001} - w_{011} - w_{100} + w_{110}$$
$$k := w_{000} - w_{001} - w_{110} + w_{111}$$
$$l := w_{010} - w_{011} - w_{100} + w_{101}$$
$$m := w_{001} + w_{010} + w_{100} - w_{111} - 2w_{000}$$
$$n := w_{011} + w_{101} + w_{110} - w_{000} - 2w_{111}$$
$$o := w_{010} + w_{100} + w_{111} - w_{001} - 2w_{110}$$
$$p := w_{000} + w_{011} + w_{101} - w_{110} - 2w_{001}$$
$$q := w_{001} + w_{100} + w_{111} - w_{010} - 2w_{101}$$
$$r := w_{000} + w_{011} + w_{110} - w_{101} - 2w_{010}$$
$$s := w_{000} + w_{101} + w_{110} - w_{011} - 2w_{100}$$
$$t := w_{001} + w_{010} + w_{111} - w_{100} - 2w_{011}$$

Before describing applications of Theorem 3 in more detail, we will compare different approaches to epistasis. As already noted, interaction coordinates (*Beerenwinkel et al., 2007*) and Walsh coefficients of order two or more (*Weinreich et al., 2013*) differ only by a scalar. However, circuits provide information of a different type. To see this, we consider the two circuits

$$a = w_{000} - w_{010} - w_{100} + w_{110}$$
$$b = w_{001} - w_{011} - w_{101} + w_{111}$$

which measure epistasis between the first and the second locus conditional on the third locus being

fixed to 0 and 1, respectively. If $a = -b$ for a system, then $u_{110} = 0$ (and the Walsh coefficient $E_{110} = 0$ as well), because $u_{110} = a + b$. If, in addition, $|a| = |b|$ is a large number, then the first and second loci have substantial interactions as measured by $a$ and $b$, yet the interaction coordinate $u_{110}$ captures only the average effect which is zero and would indicate no interaction.

Even if one knows the signs of all interaction coordinates $u_{111}$, $u_{110}$, $u_{101}$, and $u_{011}$, one may still be ignorant about important gene interactions. In contrast, the signs of all 20 circuits provide a more complete description of the gene interactions from a qualitative point of view. In this sense, it is natural to say that two fitness landscapes have similar gene interactions if their circuit sign patterns agree.

One type of circuits measures conditional epistasis, and they relate to interaction coordinates in an interesting way. More precisely, conditional epistasis concerns subsystem obtained by fixing a subset of coordinates at 0 or 1 and varying the remaining loci. Conditional epistasis for an $n$-locus subsystem agrees with the total $n$-way epistasis for the subsystem. In particular, the circuits $a$ and $b$ measure conditional epistasis. The circuit $a$ measures epistasis for the two-locus subsystem of genotypes with last coordinate 0, and the circuit $b$ measures epistasis for the two-locus subsystem of genotypes with last coordinate at 1. As mentioned, the interaction coordinate $u_{110}$ is the average of $a$ and $b$ (modulo a constant).

The relation between interaction coordinates and circuits is similar for larger systems. For instance, the circuit

$$w_{0000} - w_{0100} - w_{1000} + w_{1100},$$

measures conditional epistasis for the two-locus subsystem obtained by fixing the last two coordinates at zero. The interaction coordinate $u_{1100}$ measures the average effect of four different circuits that measure conditional epistasis. In summary, all interaction coordinates can be interpreted as averages of circuits expressing conditional epistasis.

Technically, the 20 circuits can be obtained as linear combinations of the interaction coordinates $u_{111}$, $u_{110}$, $u_{101}$, and $u_{011}$. However, none of the interaction coordinates are themselves circuits, since they do not satisfy the condition of being minimal.

The circuits $a, \ldots, f$ all measure conditional two-way epistasis between two loci when the allele at the third locus is fixed. But there are other types of circuits. The circuits $g, \ldots, l$ relate marginal epistasis of two pairs of loci, and the circuits $m, \ldots, t$ relate the three-way interaction to the total two-way epistasis (*Beerenwinkel et al., 2007*).

For a given circuit, some rank orders imply that the circuit is positive, that is, the circuit is positive for all fitness values compatible with the rank order. Similarly, some rank orders imply that the circuit is negative, whereas the sign cannot be determined from other rank orders. We will use Theorem 3 to check whether a rank order determines the sign of a circuit or not.

**Corollary 4.** *For the circuits $a, \ldots l$, two thirds of all possible rank orders determine the sign of the circuit. For the circuits $m, \ldots, t$, one half of all possible rank orders determine the sign of the circuit.*

*Proof.* Fix one of the circuits from $a$ to $l$ and a rank order. The circuit has exactly four variables with non-zero coefficients (for instance, for the circuit $a$ the variables are $w_{000}$, $w_{100}$, $w_{010}$, $w_{110}$, so that $\varphi^a$ maps rank orders to four-letter words). By Theorem 3, the rank order implies that the circuit differs from zero when it is mapped to one of the Dyck words *PPNN*, *PNPN*, *NNPP* or *NPNP* under $\varphi$, whereas the sign of the circuit is not determined when the word is *PNNP* or *NPPN*. One concludes that the sign of a given circuit from $a$ to $l$ is determined for $2/3$ of the rank orders.

Using a similar argument, we consider words of length 6 for the circuits labeled $m$ to $t$. There are in total 20 words consisting of $3\,P$'s and $3\,N$'s. Ten of them are Dyck words. We conclude that the sign of a given circuit from $m$ to $t$ is determined for $1/2$ of the rank orders. □

In general, it is not possible to decompose the word obtained for analyzing $n$-way epistasis into informative subunits. For example, as mentioned in the main text, the first half of the word *ooeeeooe* is *ooee*, and it does not appear to reveal any interesting information about the system. On the other hand, if one knows that the word *ooeeeooe* was obtained from the rank order

$$w_{010} > w_{111} > w_{110} > w_{101} > w_{011} > w_{100} > w_{001} > w_{000},$$

then one can identify meaningful parts of the word. For instance, consider the subsystem of genotypes with last coordinate 0. The corresponding letters (the first, third, sixth and eighth letter of the

**Table 2.** Comparison of the rank order method with t-test.
The first column lists the four interaction coordinates and twenty circuits. The second column shows p-values returned by Student's t-test based on fitness measurements. The third column shows which interactions are significant based on 0.03 threshold and their signs. For comparison, the last column displays the results obtained from rank order methods.

| Interaction | p-value | Result | From rank order |
|---|---|---|---|
| $u_{011}$ | 1.13e-31 | + | 0 |
| $u_{101}$ | 2.67e-12 | − | 0 |
| $u_{110}$ | 1.20e-24 | − | 0 |
| $u_{111}$ | 1.50e-29 | + | + |
| $a$ | 7.10e-16 | − | + |
| $b$ | 5.23e-32 | − | − |
| $c$ | 7.62e-04 | + | + |
| $d$ | 8.36e-68 | − | − |
| $e$ | 1.39e-38 | + | + |
| $f$ | 2.59e-01 | 0 | 0 |
| $g$ | 3.10e-59 | − | 0 |
| $h$ | 2.22e-02 | − | + |
| $i$ | 7.97e-05 | + | 0 |
| $j$ | 2.20e-32 | − | − |
| $k$ | 1.96e-05 | + | 0 |
| $l$ | 7.50e-51 | − | − |
| $m$ | 4.88e-07 | − | 0 |
| $n$ | 9.87e-37 | + | 0 |
| $o$ | 8.83e-03 | + | 0 |
| $p$ | 7.18e-19 | + | + |
| $q$ | 1.94e-01 | 0 | 0 |
| $r$ | 5.02e-50 | + | + |
| $s$ | 7.10e-27 | − | 0 |
| $t$ | 8.49e-61 | − | − |

DOI: https://doi.org/10.7554/eLife.28629.013

word *ooeeeooe*) form the four-letter word *oeoe*, which implies that the corresponding subsystem has sign epistasis. Moreover, there is no connection between words used for analyzing three-way epistasis and the words representing Walsh-coefficients. For instance, in order to analyze $u_{110}$ one maps the rank order above to *NPPNNNPP*, but the two words *ooeeeooe* and *NPPNNNPP* are unrelated.

The gene interactions for a biallelic three-locus system can be classified in terms of shapes of the fitness landscape, or triangulations of the 3-cube (*Beerenwinkel et al., 2007*). There are 74 shapes for the 3-cube. The shape of the fitness landscapes is determined by the signs of the 20 circuits. It follows that rank orders provide some information about possible shapes. However, the following result shows that rank orders do not determine shapes.

**Proposition 5.** *Consider a three-locus biallelic system. No rank order determines the shape of a fitness landscape.*

*Proof.* The result follows from the characterization of shapes for the 3-cube in (*Beerenwinkel et al., 2007*), where each shape is described in terms of a circuit sign pattern. We verified computationally that no rank order implies that all the circuits have the signs which describe a particular shape (https://github.com/gavruskin/fitlands#analysis-of-rank-orders) . More precisely, for every circuit $a, \ldots, t$, we determined the set of all rank orders that imply that the circuit is positive or negative.

For every rank order, we then considered the circuit signs determined by the order. In no case did a rank order determine all the circuit signs necessary for describing a particular shape.□

The fact that rank orders do not determine the shape of a fitness landscape over a three-locus system is not surprising. Shapes reflect interactions in a very fine scaled way, whereas rank orders provide only coarse information.

**Corollary 6.** *For each of the interaction coordinates $u_{110}$, $u_{101}$, and $u_{011}$, the number of rank orders which determines its sign is 16,128.*

*Proof.* The linear form for each interaction coordinate consists of 8 elements, 4 with positive signs and 4 with negative signs. Notice that $16,128$ rank orders imply three-way epistasis, by Proposition 1. By Theorem 3 the problem can be reduced to counting Dyck words of length 8. It follows that the number of rank orders is $16,128$ for each interaction coordinate.□

Note that the sign of a given interaction coordinate is determined for $16,128$ out of $40,320$ rank orders, that is $2/5$ or all rank orders. As mentioned, the Walsh coefficients $E_{110}$, $E_{101}$, $E_{011}$ differ from the interaction coordinates $u_{110}$, $u_{101}$, $u_{011}$ only by a scalar, so that Corollary 6 applies to the coefficients as well.

## Partial orders and fitness graphs

We now consider partial orders, for instance,

$$w_{111} > w_{110}, w_{100}, w_{010}, w_{001} > w_{000} > w_{101}, w_{011}$$

for a three-locus system. Arguing as in the proof of Proposition 1, the (unknown) total order is mapped to the word $oxxxxeee$ under $\varphi^{u_{1\ldots1}}$, where $xxxx$ is some permutation of $eooo$. For any such permutation we get a Dyck word. It follows that the system has three-way epistasis. This condition can be stated and proved in a more general form.

**Proposition 7.** *Consider an n-locus biallelic system. Let $e_i$ and $o_i$ be defined as in the proof of Proposition 1. If there exists a partition of the total set of fitness values into pairs $(e_i, o_i)$, where $e_i > o_i$ for all i, then one can conclude n-way epistasis. By symmetry, the same is true for a partition where $e_i < o_i$ for each pair.*

*Proof.* We will verify that the existence of a partition as described is equivalent to the order being mapped to a Dyck word under the map $\varphi^{u_{1\ldots1}}$. It is immediate that the existence of such a partition implies that the order is mapped to a Dyck word. Conversely, if the rank order is mapped to a Dyck word under $\varphi^{u_{1\ldots1}}$, then one can construct a partition as follows. One pair in the partition corresponds to the first $e$ and the first $o$ in the Dyck word, a second pair corresponds to the second $e$ and the second $o$ in the word, and so forth. This partition has the desired property.□

A fitness graph is a directed acyclic graph where each node represents a genotype, and arrows connect each pair of mutational neighbors, directed toward the node representing the genotype of higher fitness. Moreover, fitness graphs are structured so that the node labeled $0\ldots0$ is at the bottom, genotypes with exactly one 1 on the level above, and so forth (see *Figure 6*).

Our systematic analysis of fitness graphs takes advantage of the fact that some graphs are isomorphic, that is, there exists an edge preserving bijection between the nodes of the graphs. In biological terms, an isomorphism can be considered a relabeling of the genotypes such that mutational neighbors stay neighbors and the direction of arrows indicating higher fitness is preserved. For example, *Figure 5* shows two isomorphic fitness graphs.

The analysis of the two-locus case is straightforward. An arbitrary fitness graph is isomorphic to a graph where 00 has the lowest fitness. There are four fitness graphs satisfying the assumption. Indeed, two of the arrows point up, so that there are in total $2 \times 2 = 4$ possible fitness graphs depending on the directions of the remaining arrows (*Figure 3*). By inspection, two of the graphs in the figure are isomorphic. Consequently, there are three different fitness graphs for two-locus systems up to isomorphism.

Some fitness graphs imply epistasis, whereas other fitness graphs are compatible with additive fitness. As illustrated in the two-locus case, a fitness graph is compatible with additive fitness if all arrows point up, that is toward a higher level. More generally, a fitness graph implies epistasis unless it is isomorphic to a graph where all arrows point up. Indeed, the graph implies sign epistasis unless such an isomorphism exists. (*Weinreich et al., 2005*; *Crona et al., 2013*).

**Table 3.** Data from (*Franke et al., 2011*) on a 5-locus system determined by the mutations *fwnA1, argH12, pyrA5, leuA1*, and *pheA1*.

We consider 5-way epistasis for the system. The first column lists the ranking of the genotypes, where "?" means missing measurement. The eighth column indicates whether the genotype is odd or even. The ninth and tenth columns show the cumulative number of $o$'s and $e$'s, respectively. The last column indicates whether the number of $o$'s exceeds the number of $e$'s ($-$) or vice versa ($+$). We see that if genotype 11101 has higher fitness than genotype 10011, genotypes 11000, 10010 are ranked arbitrarily, the missing genotype 10111 has rank $1 - 15$, and 11010 rank $20 - 32$, then the last column would change to all $+$'s, so the rank order would imply $u_{11111} > 0$.

| Rank | fwn | arg | pyr | leu | phe | #mutations | $o/e$ | #$o$ cumul. | #$e$ cumul. | neg. vs pos. |
|---|---|---|---|---|---|---|---|---|---|---|
| 1 | 0 | 0 | 0 | 0 | 0 | 0 | e | 0 | 1 | + |
| 2 | 1 | 0 | 0 | 0 | 1 | 2 | e | 0 | 2 | + |
| 3 | 0 | 1 | 0 | 1 | 1 | 3 | o | 1 | 2 | + |
| 4 | 0 | 1 | 0 | 0 | 1 | 2 | e | 1 | 3 | + |
| 5 | 1 | 1 | 0 | 1 | 1 | 4 | e | 1 | 4 | + |
| 6 (or 7) | 1 | 1 | 0 | 0 | 0 | 2 | e | 1 | 5 | + |
| 7 (or 6) | 1 | 0 | 0 | 1 | 0 | 2 | e | 1 | 6 | + |
| 8 | 0 | 0 | 0 | 0 | 1 | 1 | o | 2 | 6 | + |
| 9 | 1 | 1 | 1 | 0 | 0 | 3 | o | 3 | 6 | + |
| 10 | 0 | 1 | 0 | 0 | 0 | 1 | o | 4 | 6 | + |
| 11 | 0 | 1 | 1 | 0 | 1 | 3 | o | 5 | 6 | + |
| 12 | 1 | 0 | 1 | 0 | 0 | 2 | e | 5 | 7 | + |
| 13 | 0 | 0 | 0 | 1 | 0 | 1 | o | 6 | 7 | + |
| 14 | 1 | 0 | 0 | 0 | 0 | 1 | o | 7 | 7 | + |
| 15 | 1 | 1 | 0 | 0 | 1 | 3 | o | 8 | 7 | − |
| 16 | 0 | 1 | 1 | 0 | 0 | 2 | e | 8 | 8 | + |
| 17 | 0 | 0 | 1 | 1 | 1 | 3 | o | 9 | 8 | − |
| 18 (or 19) | 1 | 1 | 1 | 0 | 1 | 4 | e | 9 | 9 | + |
| 19 (or 18) | 1 | 0 | 0 | 1 | 1 | 3 | o | 10 | 9 | − |
| 20 | 0 | 0 | 0 | 1 | 1 | 2 | e | 10 | 10 | + |
| 21 | 0 | 1 | 0 | 1 | 0 | 2 | e | 10 | 11 | + |
| 22 | 0 | 0 | 1 | 0 | 1 | 2 | e | 10 | 12 | + |
| 23 | 0 | 0 | 1 | 0 | 0 | 1 | o | 11 | 12 | + |
| 24 | 0 | 1 | 1 | 1 | 1 | 4 | e | 11 | 13 | + |
| 25 | 0 | 1 | 1 | 1 | 0 | 3 | o | 12 | 13 | + |
| 26 | 0 | 0 | 1 | 1 | 0 | 2 | e | 12 | 14 | + |
| 27 | 1 | 0 | 1 | 0 | 1 | 3 | o | 13 | 14 | + |
| 28 | 1 | 1 | 1 | 1 | 0 | 4 | e | 13 | 15 | + |
| 29 | 1 | 1 | 1 | 1 | 1 | 5 | o | 14 | 15 | + |
| 30 | 1 | 0 | 1 | 1 | 0 | 3 | o | 15 | 15 | + |
| ? | 1 | 1 | 0 | 1 | 0 | 3 | o | ? | ? | ? |
| ? | 1 | 0 | 1 | 1 | 1 | 4 | e | ? | ? | ? |

DOI: https://doi.org/10.7554/eLife.28629.014

Accordingly, we can characterize rank orders that are compatible with fitness graphs with all arrows up. After relabeling genotypes, we can assume that the genotype $0\ldots0$ has the lowest fitness in the system. Then a rank order is compatible with a fitness graph with all arrows up, exactly if for each genotype replacing 0 by 1 results in higher fitness. For instance, $w_{00} < w_{10} < w_{01} < w_{11}$ is compatible with such a graph whereas the rank order $w_{00} < w_{11} < w_{01} < w_{10}$ is not.

If 000 has the lowest fitness in the system, one can verify that exactly 48 rank orders are compatible with the fitness graph with all arrows up. It follows that in total $8 \times 48 = 384$ rank orders are compatible with such graphs since there are eight genotypes in a three-locus system, each of which can have lowest fitness.

Interestingly, the same result can be obtained from theory on house-of-cards landscapes. Since all rank orders are equally likely under this statistical fitness landscape model, the fraction of rank orders that imply sign epistasis agrees with the probability of sign epistasis. This probability is $\frac{104}{105}$ (Schmiegelt and Krug 2014, *Table 2*). It follows that $8! \times \frac{1}{105} = 384$ rank orders are compatible with fitness graphs with all arrows up, that is, the total number of rank orders multiplied by the fraction of rank orders compatible with such graphs.

As we have already seen, rank orders have potential far beyond detecting whether or not there is epistasis in a system. The same is true for fitness graphs, and we proceed with higher order interactions. In order to analyze fitness graphs and three-way epistasis, we consider the set of rank orders compatible with a given fitness graphs. For instance, the fitness graph in *Figure 10* is compatible with the following two rank orders:

$$w_{111} > w_{000} > w_{100} > w_{010} > w_{001} > w_{110} > w_{101} > w_{011}, \tag{5}$$

$$w_{000} > w_{111} > w_{100} > w_{010} > w_{001} > w_{110} > w_{101} > w_{011}. \tag{6}$$

The first order implies three-way epistasis (it is mapped to *oeoooeee* under $\varphi^{u_1,\ldots,1}$) and the second does not (it is mapped to *eoooeee* under $\varphi^{u_1,\ldots,1}$). We conclude that in this case, the fitness graph does not imply higher order epistasis. However, if every rank order compatible with the fitness graph implies higher order epistasis, then the fitness graph itself does imply higher order epistasis. More generally, the same observation holds for any partial order.

**Remark.** *A partial order implies higher order epistasis exactly if all its total extensions imply higher order epistasis.*

Indeed, if all total extensions imply higher order epistasis, then in particular the (unknown) rank order does. The converse holds by definition.

Consequently, one can in principle determine if a fitness graph implies higher order epistasis by checking all of the compatible rank orders. For a systematic study of the three-locus case, it is convenient to reduce the problem to isomorphic graphs. *Figure 5* shows two isomorphic fitness graphs.

As mentioned in the main text, there are in total $1,862$ fitness graphs for three-locus systems, and the number of fitness graphs that imply higher order epistasis is 698 (37 percent). Up to isomorphism there are in total 54 fitness graphs, and 20 graphs imply higher order epistasis. This result was verified by reducing the study of all $1,862$ graphs for three-locus systems to a non-redundant list of 54 graphs, such that no two graphs in the list are isomorphic. The fact that the total number of graphs is $1,862$ follows from general theory on acyclic graphs (*Stanley, 2006*), or can be verified computationally (see below).

Isomorphisms between three-locus systems have a geometric interpretation. The fitness graph can be regarded as a three-dimensional cube, with vertices corresponding to genotypes and edges corresponding to arrows. The group of isomorphisms (see below) then corresponds to the symmetry group of the three dimensional cube (*Coxeter, 1973*). Indeed, it was by way of this equivalence that we carried out the enumeration described above.

For clarity, we give a more explicit description of the cube isomorphisms. Any isomorphism can be constructed as a composition of the following two transformations: (i) interchange of labels of the pair of alleles at a locus, and (ii) change of order of the loci in the bitstring representation of a genotype. There are in total forty-eight isomorphisms of the cube, including the identity transformation which leaves the cube unchanged. *Figure 5* is an example of such a transformation, where at the first locus the labels 0 and 1 have been swapped, and the second and third loci have been interchanged. The code used for verifying the isomorphisms is available at https://github.com/devingreene/3-cube-partial-order-count.git.

A cube has 12 edges, so that the total number graphs on the cube (graphs similar to fitness graphs, but cycles are allowed) is $2^{12} = 4096$. After exclusion of graphs with cycles, $1,862$ graphs remain. An arbitrary graph is isomorphic to a graph where 000 has the lowest fitness. This relabeling

reduces the number of graphs one has to consider substantially, as in the two-locus case. The list of remaining graphs can be reduced further using the isomorphisms described above to finally obtain 54 graphs (*Figure 3*).

### Graph theoretical aspects

As mentioned in the main text, the 20 graphs which imply higher order epistasis (see *Figure 6*) constitute a diverse category. We analyzed the category from a graph theoretical point of view, but could not see that the graphs have any property which singles them out.

Recall that a unique sink orientation is a graph where each face has no more than one sink. Equivalently, there is no subsystems with reciprocal sign epistasis (*Crona et al., 2013*; *Poelwijk et al., 2007*). The category of 20 graphs includes unique sink orientations (also called USO or AOF graphs), as well as non-USO's. Moreover, in the terminology of *Gärtner and Kaibel, 1998*, the category includes separable and non-separable graphs, as well as realizable and non-realizable graphs.

There were some indications of higher complexity for the category, but only in a statistical sense. Indeed, as can be verified from *Figure 6*, the graphs in the category have on average 1.8 sinks (a sink corresponds to a peak in the landscape), whereas the average number of sinks for all graphs is 1.6. Moreover, 5 out of the 20 graphs (25 percent) in the category are unique sink orientations, whereas in total 19 out of the 54 graphs (35 percent) are unique sink orientations.

Even though the category of fitness graphs which implies epistasis is diverse, it is still possible that a characterization exists. This is an open problem.

## Software and HIV-1 study

We have implemented algorithms based on our theoretical results in an open source software package (https://github.com/gavruskin/fitlands#fitlands). The package provides software for detecting gene interactions as described in the main text for two- and three-locus systems. Furthermore, algorithms for detecting total $n$-way epistasis, three- and four-way interaction coordinates and three-way circuit interactions have been implemented. The documentation also explains how to reproduce results for our application to HIV-1 data described in the main text.

The results of Student's t-test explained in the main text are summarized in *Table 1*, and for related code see https://github.com/gavruskin/fitlands/blob/master/HIV_2007_conventional_analysis.ipynb

## Acknowledgements

KC would like to thank Mr Dong Zhou for his study on published malaria data. We thank the reviewers for their comments, in particular regarding house-of-cards landscapes. This work was done in part while the authors were visiting the Simons Institute for the Theory of Computing.

## Additional information

### Funding
No external funding was received for this work

### Author contributions
Kristina Crona, Conceptualization, Formal analysis, Investigation, Visualization, Methodology, Writing—original draft, Project administration, Writing—review and editing, Project design; Mathematical proofs; Alex Gavryushkin, Devin Greene, Conceptualization, Resources, Data curation, Software, Formal analysis, Investigation, Visualization, Methodology, Writing—original draft, Writing—review and editing; Niko Beerenwinkel, Conceptualization, Supervision, Funding acquisition, Investigation, Methodology, Writing—original draft, Project administration, Writing—review and editing

### Author ORCIDs
Kristina Crona (iD) https://orcid.org/0000-0003-1819-474X
Alex Gavryushkin (iD) https://orcid.org/0000-0001-6299-8249

Devin Greene [iD] http://orcid.org/0000-0002-4889-9225
Niko Beerenwinkel [iD] https://orcid.org/0000-0002-0573-6119

**Decision letter and Author response**
Decision letter https://doi.org/10.7554/eLife.28629.017
Author response https://doi.org/10.7554/eLife.28629.018

## Additional files

**Supplementary files**
• Transparent reporting form
DOI: https://doi.org/10.7554/eLife.28629.015

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
