## [Decision Letter]

Thank you for submitting your article "Inferring Genetic Interactions From Comparative Fitness Data" for consideration by *eLife*. Your article has been favorably evaluated by Aviv Regev (Senior Editor) and three reviewers, one of whom, Dan Weinreich (Reviewer #1), is a member of our Board of Reviewing Editors. The following individuals involved in review of your submission have agreed to reveal their identity: Joachim Krug (Reviewer #2).

The reviewers have discussed the reviews with one another and the Reviewing Editor has drafted this decision to help you prepare a revised submission.

Summary:

This is an important contribution to a very, very challenging problem: how to assess fitness interactions (epistasis) from data. The tremendous challenge derives from the exponentially large size of the problem: to fully characterize all epistasis among n mutations requires experimental measurements for all genotype defined by every combination of the mutations. And this number grows exponentially in n. Here, the authors demonstrate that much can be learned from much smaller datasets containing information only on the sign of a subset of the differences in fitness among genotypes. Their fundamental result depends on a new connection between this rank order information and a mathematical object called a Dyck word. Apart from the practical utility of the formalism presented in this manuscript, it also provides a new perspective on the geometry of high-dimensional genotypic fitness landscapes. This is thus a very innovative contribution to a highly active field with a broad potential significance.

However, several substantial concerns have been raised that must be resolved before the manuscript can be accepted.

Essential revisions:

1) Quite importantly, you appear to have misunderstood the concept of sign epistasis as it was introduced in 2005 by Weinreich, Watson and Chao. As a consequence, you underestimate the significance of this concept for your own approach and fail to properly place your work into context. You claim twice (Introduction, tenth paragraph and Discussion, sixth paragraph) that the concept of sign epistasis "refers to the two-locus system" and that your approach generalizes this concept to any number of loci. This is incorrect. Sign epistasis means that the (fitness) effect of a mutation at a locus can be positive or negative depending on the state of other loci (the genetic background), and this definition applies in full generality to any number of loci. Moreover, the definition evidently only relies on the rank order of genotype fitness values. It thus seems that this implies that sign epistatic interactions are exactly those that can, in principle, be inferred from rank order information: In the presence of sign epistasis the rank order is not consistent with an additive landscape.

This is evident in the two-locus example, where the red and blue rank orders in Figure 1 are precisely those that display sign epistasis. It is well known from studies of 'house-of-cards' fitness landscapes (where fitness values are independent, identically distributed random variables) that the probability for a pair of loci to display sign epistasis is 2/3 [see e.g. Schmiegelt and Krug, J. Stat. Phys. 2014]. Since all rank orders are equally likely under this model, this is then also the fraction of rank orders that imply sign epistasis. For three loci the fraction of sign-epistatic landscapes is already 104/105 [this follows from Table 2 of the paper by Schmiegelt and Krug: For sign-epistatic landscapes the number of accessible paths is less than 6, which is true with probability 104/105]. If the conjecture formulated above is correct, it follows that only 40320/105=384 rank orders are not informative about any kind of epistatic interaction (3-way epistasis, circuits or interaction coordinates). This should be easy for you to check.

Related to this, the discussion of fitness graphs that are compatible with additive fitness in the subsection “1.3. Partial orders and fitness graphs” of Mathematical framework and proofs is somewhat misleading, as it gives the impression that this characterization was first derived by Crona et al. 2013. In fact, the proof that all paths to the global maximum are accessible in a landscape without sign epistasis (which implies that all arrows in the fitness graph point up) was first given by Weinreich et al. 2005.

2) Another critical point comes in the subsection “Three-locus case”: given the Dyck word for a 3-bit rank ordering, how does one pick out the three 2-bit Dyck words to assess the presence of each of the pairwise epistases? The information must be present in the longer word, and this point seems to be a tremendous, missed opportunity. And of course, such an algorithm for dissecting sub-Dyck words should be generalized to the n-locus case. After all, current thinking on higher-order epistasis focuses not only on the highest-order component but all 2^n – n – 1 components. Such dissections should also be illustrated with the empirical data analyzed here.

3) Another commonly used framework for the characterization of fitness landscapes is the Fourier-Walsh decomposition. Again, the relevance of this framework for your approach is evident – e.g., the criterion for n-way epistasis simply implies that the highest order Walsh coefficient is nonzero; the 'interaction coordinates' are proportional to Walsh coefficients; and it seems (but perhaps not) that also the 'circuits' are essentially linear combinations of Walsh coefficients. Nevertheless, the precise relation between the two approaches remains obscure to the reader. Walsh coefficients are mentioned in the main text as if the reader were assumed to know about them, but when they finally are defined on the first page of the Materials and methods, the term does not even appear. You need to explain the concept and its relation to your approach in the main text; the repeated references to Beerenwinkel et al. 2007 are not sufficient in this respect.

(Incidentally, that the criterion for n-way epistasis simply implies that the nth-order Walsh coefficient is nonzero may suggest a direct and elegant path to dissect out the lower-order effects – see the previous point.)

4) The discussion related to fitness proxies in the Introduction is somewhat muddled and confusing. Clearly the analysis presented can be applied to any scalar trait, not just fitness, and for any such trait the method will be able to detect only those components of epistasis that are invariant under monotonic transformations (which preserve rank ordering; indeed, you acknowledge this point in the second-to-last paragraph of your Conclusions). In the terminology of Weinreich et al. 2005, the method can detect sign epistasis but not magnitude epistasis. Thus for example, in the fourth paragraph of the Introduction is a statement about non-linearity in the phenotype->fitness map, not a statement about epistasis per se. This should be corrected. See also the eighth paragraph of the Introduction.

To what extent magnitude epistasis (which may of course contribute to Walsh coefficients of arbitrary order and hence also give rise to n-way epistasis) can be removed by the choice of a suitable nonlinear scale and/or of a suitable fitness proxy is a separate issue that has recently been addressed systematically by Sailer and Harms in the cited Genetics paper (see also Szendro et al., JSTAT 2013 and Neidhart, Szendro, Krug, JTB 2013 for similar discussion and evidence for high-order epistasis). Introducing the distinction between sign and magnitude epistasis early on in the Introduction would help to clarify this point.

And perhaps of related interest here, Knies and Weinreich 2017 Molecular Biology and Evolution 34(5):1040-1054 demonstrate that comparatively small epistatic terms can still induce sufficient sign epistasis to dramatically reduce the number of selectively accessible trajectories.

5) Subsection “General n-locus case”, last paragraph: as written one is left with the sense that the github implementation can only handle n = 3. Is that correct? To be useful to the community it should work for arbitrary n (up to run-time consequences). Please clarify, and if there's a good reason why the code can't handle arbitrary n, please explain. This would be a very serious limitation to the importance of the work.

6) Finally, a consolidated and more complete development of the utility of the work for the community of end users should be added. The field has probably moved beyond the simple question "Is there higher-order epistasis?" but for example, can these methods help focus the experimentalist on particular subsets of mutations that exhibit anomalous interactions for further mechanistic attention? (The holy grail would of course be if you could offer even some speculation or intuition for how rank orders could contribute to the thorny problem of capturing epistasis in GWAS studies.) Some material in this vein is presently provided in the –tenth paragraph of the Introduction and in the fourth paragraph of the subsection “Analysis of empirical fitness data”, but a tremendous opportunity to more fully capitalize on the breadth of readership remains.

This line of thinking may also suggest a modified and more broadly engaging title.

[Editors' note: further revisions were requested prior to acceptance, as described below.]

Thank you for resubmitting your work entitled "Inferring Genetic Interactions From Comparative Fitness Data" for further consideration at *eLife*. Your revised article has been evaluated by Aviv Regev as the Senior Editor and Dan Weinreich as the Reviewing Editor.

The manuscript has been improved but there are some remaining issues that need to be addressed before acceptance. Chiefly, we are not yet entirely satisfied with the response to Essential revision 1: some of the statements regarding the relation between the established concept of sign epistasis and the framework presented here are, at least, misleading. In the tenth paragraph of the Introduction you say that yours is "a more general approach" compared to the concept of sign epistasis. This is misleading, because any instance of epistasis captured by your approach is also an instance of sign epistasis; in this sense the new approach is a refinement rather than a generalization. This is clearly stated as: "Importantly, if a rank order implies that the sign of an additive dependence relation, such as a circuit or an interaction coordinate, is determined, then the system has sign epistasis." This key statement should be made (in a suitable form) in the Introduction. Similarly, you say that Weinreich et al. (2005) introduced the term sign epistasis "for bialellic two-locus systems", whereas in fact this work considered arbitrary multilocus systems. In the following sentence you claim that sign epistasis "is a special case of rank order-induced epistasis", which is again incorrect, because any case of rank order-induced epistasis implies sign epistasis for at least one pair of loci.

---

## [Author Response]

Essential revisions:1) Quite importantly, you appear to have misunderstood the concept of sign epistasis as it was introduced in 2005 by Weinreich, Watson and Chao. As a consequence, you underestimate the significance of this concept for your own approach and fail to properly place your work into context. You claim twice (Introduction, tenth paragraph and Discussion, sixth paragraph) that the concept of sign epistasis "refers to the two-locus system" and that your approach generalizes this concept to any number of loci. This is incorrect. Sign epistasis means that the (fitness) effect of a mutation at a locus can be positive or negative depending on the state of other loci (the genetic background), and this definition applies in full generality to any number of loci. Moreover, the definition evidently only relies on the rank order of genotype fitness values. It thus seems that this implies that sign epistatic interactions are exactly those that can, in principle, be inferred from rank order information: In the presence of sign epistasis the rank order is not consistent with an additive landscape.[...]Related to this, the discussion of fitness graphs that are compatible with additive fitness in the subsection “1.3. Partial orders and fitness graphs” of Mathematical framework and proofs is somewhat misleading, as it gives the impression that this characterization was first derived by Crona et al. 2013. In fact, the proof that all paths to the global maximum are accessible in a landscape without sign epistasis (which implies that all arrows in the fitness graph point up) was first given by Weinreich et al. 2005.

We recognize that the concept of sign epistasis applies to any number of loci. However, sign epistasis for a general system can be identified from 2-locus subsystems, whereas in our approach, the smallest meaningful subsystems are usually larger. In that sense we have extended the perspective to any number of loci. In fact, any epistasis concept that can be expressed in terms of a linear equation has a signed, i.e., a rank order-induced, correspondence. One can ask if a system has signed n-way epistasis, signed circuits, or signed Walsh coefficients. Clearly, sign epistasis was the first concept of this type and we have clarified and emphasized the connection in the Introduction, Results, and Discussion.

We are very grateful for the comment on the 384 rank orders that are not informative about any kind of epistatic interaction based on the Schmiegelt and Krug (2014) paper. We have included this observation in the last paragraph of the subsection “Two-locus case” and the argument in Section 1.3 of the Materials and methods.

We have also added the Weinreich et al. (2005) reference to the discussion on additive fitness and fitness graphs as suggested (Materials and methods, subsection “1.3. Partial orders and fitness graphs”, seventh paragraph).

2) Another critical point comes in the subsection “Three-locus case”: given the Dyck word for a 3-bit rank ordering, how does one pick out the three 2-bit Dyck words to assess the presence of each of the pairwise epistases? The information must be present in the longer word, and this point seems to be a tremendous, missed opportunity. And of course, such an algorithm for dissecting sub-Dyck words should be generalized to the n-locus case. After all, current thinking on higher-order epistasis focuses not only on the highest-order component but all 2^n – n – 1 components. Such dissections should also be illustrated with the empirical data analyzed here.

This is an interesting question. Unfortunately, the substrings of the word under consideration do, in general, not reveal information about interactions in the corresponding subsystem. Similarly, there is no obvious connection between the word used for analyzing three-way epistasis and the words representing Walsh-coefficients. We illustrate this point by way of a small example in the Results (subsection “General n-locus case”, seventh paragraph) and Materials and methods.

3) Another commonly used framework for the characterization of fitness landscapes is the Fourier-Walsh decomposition. Again, the relevance of this framework for your approach is evident – e.g., the criterion for n-way epistasis simply implies that the highest order Walsh coefficient is nonzero; the 'interaction coordinates' are proportional to Walsh coefficients; and it seems (but perhaps not) that also the 'circuits' are essentially linear combinations of Walsh coefficients. Nevertheless, the precise relation between the two approaches remains obscure to the reader. Walsh coefficients are mentioned in the main text as if the reader were assumed to know about them, but when they finally are defined on the first page of the Materials and methods, the term does not even appear. You need to explain the concept and its relation to your approach in the main text; the repeated references to Beerenwinkel et al. 2007 are not sufficient in this respect.(Incidentally, that the criterion for n-way epistasis simply implies that the nth-order Walsh coefficient is nonzero may suggest a direct and elegant path to dissect out the lower-order effects – see the previous point.)

We have clarified the relation between interaction coordinates and Walsh coefficients in the fourth paragraph of Mathematical framework and proofs. As we only care about the sign of (any type of) interaction here, all statements about interaction coordinates hold for the corresponding Walsh coefficients as well. We did not see a direct path to dissect lower-order effects in this manner. We have also added a more detailed discussion on interaction coordinates and Walsh coefficients versus circuits in the Results, subsection “General n-locus case”, and Mathematical framework and proofs, subsection “1.2. Circuits”.

4) The discussion related to fitness proxies in the Introduction is somewhat muddled and confusing. Clearly the analysis presented can be applied to any scalar trait, not just fitness, and for any such trait the method will be able to detect only those components of epistasis that are invariant under monotonic transformations (which preserve rank ordering; indeed, you acknowledge this point in the second-to-last paragraph of your Conclusions). In the terminology of Weinreich et al. 2005, the method can detect sign epistasis but not magnitude epistasis. Thus for example, in the fourth paragraph of the Introduction is a statement about non-linearity in the phenotype->fitness map, not a statement about epistasis per se. This should be corrected. See also the eighth paragraph of the Introduction.[…]And perhaps of related interest here, Knies and Weinreich 2017 Molecular Biology and Ev1olution 34(5):1040-1054 demonstrate that comparatively small epistatic terms can still induce sufficient sign epistasis to dramatically reduce the number of selectively accessible trajectories.

Thank you for mentioning these additional references, which we have integrated into the Introduction, seventh paragraph. The sentence “Although it is possible to study epistasis of the proxy phenotype, additive proxy data does not in general imply absence of epistasis with respect to fitness” does not assume a phenotype-fitness map. It only says that from no epistasis in the proxy landscape one can, in general, not conclude no epistasis in the fitness landscape. To clarify this, we have removed the previous sentence and have rewritten the sentence above as follows: “Although it is possible to study epistasis of the proxy data, in general, presence or absence of epistasis in the proxy landscape does not imply presence or absence of epistasis in the fitness landscape.” avoiding the term “proxy phenotype”.

5) Subsection “General n-locus case”, last paragraph: as written one is left with the sense that the github implementation can only handle n = 3. Is that correct? To be useful to the community it should work for arbitrary n (up to run-time consequences). Please clarify, and if there's a good reason why the code can't handle arbitrary n, please explain. This would be a very serious limitation to the importance of the work.

Our implementation can handle the case of arbitrary *n*. We have rewritten the text accordingly (subsection “General *n*-locus case”, last paragraph). The GitHub repository contains a detailed technical description (https://github.com/gavruskin/fitlands/blob/master/Four-way_interaction_coordinates_and_total_n-way_interaction.ipynb).

6) Finally, a consolidated and more complete development of the utility of the work for the community of end users should be added. The field has probably moved beyond the simple question "Is there higher-order epistasis?" but for example, can these methods help focus the experimentalist on particular subsets of mutations that exhibit anomalous interactions for further mechanistic attention? (The holy grail would of course be if you could offer even some speculation or intuition for how rank orders could contribute to the thorny problem of capturing epistasis in GWAS studies.) Some material in this vein is presently provided in the –tenth paragraph of the Introduction and in the fourth paragraph of the subsection “Analysis of empirical fitness data”, but a tremendous opportunity to more fully capitalize on the breadth of readership remains.This line of thinking may also suggest a modified and more broadly engaging title.

We have added a discussion on how our methods may contribute to detecting epistasis in GWAS (Discussion, eighth paragraph). We do not believe that these remarks justify a more broadly engaging title and would prefer to leave it unchanged.

[Editors' note: further revisions were requested prior to acceptance, as described below.]

The manuscript has been improved but there are some remaining issues that need to be addressed before acceptance. Chiefly, we are not yet entirely satisfied with the response to Essential revision 1: some of the statements regarding the relation between the established concept of sign epistasis and the framework presented here are, at least, misleading. In the tenth paragraph of the Introduction you say that yours is "a more general approach" compared to the concept of sign epistasis. This is misleading, because any instance of epistasis captured by your approach is also an instance of sign epistasis; in this sense the new approach is a refinement rather than a generalization. This is clearly stated as: "Importantly, if a rank order implies that the sign of an additive dependence relation, such as a circuit or an interaction coordinate, is determined, then the system has sign epistasis." This key statement should be made (in a suitable form) in the Introduction. Similarly, you say that Weinreich et al. (2005) introduced the term sign epistasis "for bialellic two-locus systems", whereas in fact this work considered arbitrary multilocus systems. In the following sentence you claim that sign epistasis "is a special case of rank order-induced epistasis", which is again incorrect, because any case of rank order-induced epistasis implies sign epistasis for at least one pair of loci.

Indeed, rank order induced interactions, or signed interactions, are conceptually related to sign epistasis. There is a potential for theoretical advances on signed interactions, similar to published results on global peaks and sign epistasis. We have emphasized these facts more in the revised version. However, the exact relationship between the concept of sign epistasis and the class of signed interactions we consider here is rather involved, but the detailed exposition of this relationship goes beyond the scope of this paper and will be explored in future work.

Actions:

The manuscript has been revised accordingly. We have also removed the text that can be misinterpreted regarding sign epistasis and two-locus system. Specifically, the following changes have been made:

1) New text:

“Here, we develop a related approach based on rank orders that applies to higher order epistasis as well as other measures of gene interactions. […] The theory of sign epistasis stands as a model for development in the area, and there is a potential for understanding global properties of fitness landscapes in terms of (local) signed interactions, similar to results for sign epistasis.”

Deleted text:

“Here, we develop a more general approach that applies to higher order epistasis as well as other measures of gene interactions. […] Similarly, one can consider signed versions of other types of gene interactions.”

2) Deleted text:

“Importantly, if a rank order implies that the sign of an additive dependence relation, such as a circuit or an interaction coordinates, is determined, then the system has sign epistasis.”

3) New text:

“This distinction was pointed out by Weinreich, Watson, and Chao (2005) who introduced the term sign epistasis. […] Rank order-induced gene interactions of any type can thus be regarded as analogues to sign epistasis.”

Deleted text:

“This distinction was pointed out for biallelic two-locus systems by Weinreich, Watson, and Chao (2005) who introduced the term sign epistasis. […] Rank order-induced gene interactions of any type can thus be regarded as analogues to sign epistasis, and this new concept is meaningful in a broader context.”

4) New text:

“A very useful feature of sign epistasis is that one can identify, or rule out, sign epistasis in a system from inspection of two-locus subsystems only.”

Deleted text:

“A very useful feature of sign epistasis is that one can identify, or rule out, sign epistasis from inspection of two-locus subsystems only.”

5) Further revisions related to this issue:

New text:

“If the genotype 00 has minimal fitness in the system, then rank orders are compatible with graph (a) exactly if they satisfy this property. […] In particular, only 384 out of (2^3^)! = 40,320 rank orders are compatible with fitness graphs with all arrows up (after relabeling) for the three-locus system, which we consider next (Meterials and methods, Section 1.3).”

Several changes have been made after “More generally, a fitness graph implies epistasis unless it is isomorphic to a graph with all arrows up”. In most cases “additive fitness” has been replace by “fitness graphs with all arrows up”, or similarly.